

# Spatial and temporal variability in soil and vegetation carbon dynamics under experimental drought and soil amendments

Daniela Guasconi[1], Sara A. O. Cousins[1], Stefano Manzoni[1], Nina Roth[1], Gustaf Hugelius[1]

1Department of Physical Geography and Bolin Centre for Climate Research, Stockholm University, Stockholm, Sweden

*Correspondence to*: Daniela Guasconi (daniela.guasconi@natgeo.su.se)

## Abstract

Soils are the largest carbon (C) pool on the planet, and grassland soils have a particularly large C sequestration potential. Appropriate land management strategies, such as organic matter additions, can improve soil health, increase soil C stocks, and increase grassland resilience to drought by improving soil moisture retention. However, soil C dynamics are deeply linked to vegetation response to changes in both management and climate, which may also be manifested differently in roots and shoots. This study presents findings from a three-year experiment that assessed the impact of a compost amendment and of reduced precipitation on soil and vegetation C pools. Compost addition increased aboveground biomass and soil C content (%C), but because bulk density decreased, there was no significant effect on soil C stocks. Drought decreased aboveground biomass, but did not significantly affect root biomass. Overall, the soil amendment shifted C allocation to aboveground plant organs, and drought to belowground organs. We also observed significant spatial and temporal variability in vegetation biomass and soil C over the study period. These results highlight the need to consider multiple biotic and abiotic factors driving ecosystem C dynamics across spatial scales when upscaling results from field trials.

## Introduction

Management of soil health and soil carbon (C) stocks has been receiving increasing attention in the past years, with growing awareness that soils provide vital ecosystem services and act as C sinks. Concerns about soil erosion and historic soil C depletion in agricultural and grassland soils (Sanderman et al. 2017, Bai and Cotrufo, 2022) have motivated the development of sustainable land management strategies, sometimes named "carbon farming". One such strategy is the use of soil C amendments (Ryals and Silver 2013; Ryals et al. 2015; Keesstra et al. 2016; Fischer et al. 2019;



Garbowski et al. 2023), including compost, biochar, and various types of manure. These treatments
can be applied on croplands or rangelands, in single or multiple applications, and can increase soil
aggregation (Sarker et al. 2022) and mitigate soil organic carbon (SOC) loss resulting from human
activities such as tilling. In some cases, C amendments have even been proposed as means to actively
sequester C in soils, with initiatives like the "4 per 1000" (Minasny et al. 2017) promoting their
implementation as a climate change mitigation strategy. Consequently, soil C management methods
aim to shift the ecosystem C balance by facilitating the movement of C from the atmosphere into
vegetation and subsequently, into the soil C pool, where it can be retained over long time scales.
Adding C to soils in the form of amendments directly increases the standing stock of SOC, but C
amendments may also act as a primer to an ecosystems natural ability to sequester C via indirect
effects. If C amendments promote plant growth, they also increase the natural rate of C input to the
soil and thus potentially SOC stocks if the additional C input is stabilized and does not promote
mineralization of native SOC. Considering these indirect effects requires an ecosystem-level
perspective on soil C sequestration potential that accounts for both below- and above-ground
vegetation contributions. Root biomass, belowground plant organs and root exudates are an integral
part of soil C formation and retention (Jackson et al., 2017). However, aboveground plant biomass
should also be included in these assessments to identify potential trade-offs in above-and
belowground C allocation within the vegetation pool, and to determine whether there are changes in
vegetation C pools which may affect the soil C pool. It is especially important to determine the
proportion of plant litter that contributes to soil organic matter (SOM) formation and stabilization
(Cotrufo et al. 2015). It has been argued that an approach that accounts for above- and belowground
interactions is essential to get a comprehensive understanding of ecosystem C dynamics (Heimann
and Reichstein, 2008).
Land management practices, whether through conventional or regenerative methods, can significantly
impact soil C and plant biomass both above- and belowground. For instance, historic land use has
depleted the soil of C (Sanderman et al. 2017), and agricultural lands have lower root biomass C
compared to managed grasslands (Beniston et al. 2014). Organic amendments affect different
properties and mechanisms in soils, including soil aggregation and structure, soil microbial
communities, and plant roots, and the interactions between these elements (Sarker et al. 2022, Liu et



al. 2016). Amendments provide nutrients that can stimulate microbial activity and plant growth (Hammerschmiedt et al. 2021), and increase crop yields (Luo et al. 2018). Another promising application of soil organic amendments is their use to mitigate the negative effects of drought on vegetation and soil microbial communities (Fischer et al. 2019). Future climate projections indicate an increase in extreme weather events, including longer and more frequent droughts (IPCC, 2021), which may decrease the soil C sequestration capacity of grasslands by altering plant community composition, plant C allocation and microbial processes (Bai and Cotrufo, 2022). Organic soil amendments can enhance resilience to drought by increasing soil capacity to retain soil moisture (Fischer et al. 2019; Haque et al. 2021). This can also indirectly benefit the ecosystem C balance by partly compensating the drought-induced loss of plant biomass (Kallenbach et al. 2019; Ali et al. 2017).

Many studies have investigated the effects of organic amendments on aboveground biomass (Ryals et al. 2016) and on crop yield (Luo et al. 2018; Ahmad et al. 2009, Hirte et al. 2021), but fewer focused on roots and on non-cultivated grasslands. Garbowski et al. (2020) observed that soil amendments can have an effect only on aboveground biomass or only on belowground biomass, which may be expected since roots and shoots respond differently to changes in nutrient (Hayes et al. 2017) and water availability (Wilcox et al. 2017; Guasconi et al. 2023). Furthermore, soil and plant communities can show great variability in response to both drought (Guasconi et al. 2023; Canarini et al. 2017) and soil amendments (Gebhardt et al. 2017). This variability derives partly from the variable physical properties of soil, but can also depend on land use history or on small- and large-scale topography (Wang et al. 2020). This may partly explain why results from field vs. lab experiments can differ considerably (Canarini et al. 2017), and highlights the need for more field-based data collections—in particular under experimental conditions that combine soil amendments and drought.

Here, we present the results of a field experiment designed to assess the effects of a soil amendment and of reduced precipitation on both soil and vegetation C pools, where we observed changes at various soil depths, in two grasslands, and at two catenary positions. Because the effects of already partly decomposed organic amendments can be expected to be longer-lasting than those of easily decomposable ones (Sarker et al. 2022), we applied a one-time compost treatment coupled to a yearly growing season drought and investigated their effect on the soil C stocks after three full growing



seasons. We measured soil organic C contents (mass of C per unit mass of soil) and soil C stocks
(calculated as C content × bulk density × layer thickness) at different depths within the soil profile,
and vegetation biomass (encompassing both root biomass and aboveground biomass, including plant
litter). We tested the hypotheses that:
1) compost amendment increases soil C content and plant growth (both having positive effects on C
stocks), while decreasing soil bulk density (having a negative effect on C stocks); we expect that these
mechanisms have counteracting effects on net soil C storage;
2) by decreasing both productivity (organic C input) and respiration (microbial decomposition of
SOM), drought will have a weak or non-detectable effect on SOC;
3) compost amendment mitigates the loss of soil moisture under drought which may alleviate loss of
plant growth under drought.
Because of the sensitivity of vegetation to natural variability in precipitation (Liu et al. 2020) and
potential effects of landscape heterogeneity on both soil C dynamics and plant growth (Sharma et al.
2022; Guo et al. 2018), the analyses include testing for differences in the control plots between the
start and the end of the experiment, as well quantifying the variability given by grassland and catenary
position, which we expect might lead to variations in all C pools.

## Methods

### Site description and experimental setup

The experimental site was established in summer 2019 in the proximity of Tovetorp Research Station
south of Stockholm, Sweden, and consists of two former arable fields (hereafter called "Tovetorp"
and "Ämtvik"), each with an upper and a lower catenary position. Today the fields are managed for
grazing and haymaking. In each of these four locations, four treatments (compost, drought, drought-
compost, control) were applied in three replicates, resulting in 12 plots per location and 48 plots in
total. Each plot measured 2x2 m. Soil in all locations is rich in clay and ranges from silty clay to silty
loam. The compost was made of *Zea mays* with a C:N ratio of 9.8 and $\delta^{13}$C value of about -15.39‰
and was applied in mid-February 2020 as a thin surface layer of ca. 11 kg per m$^2$ (wet weight), similar
to the procedure described in Ryals and Silver (2013). The total amount of C added is estimated to be
~0.54 kg C m$^{-2}$. The $\delta^{13}$C isotope ratio of the compost is higher than that of bulk soil, which means
that the $\delta^{13}$C isotope ratios of different treatments can be used to assess if and where in the soil the





compost material is retained after the three years of treatment. The drought treatment followed the
guidelines of the Drought-Net Research Coordination Network (Knapp et al. 2017; Yahdjian and
Sala, 2002), and consisted of 12 rainout shelters (3 per location) with roofs made out of evenly-placed
v-shaped polycarbonate strips designed to exclude 60% of the precipitation during the entire growing
season (in place from beginning of July to end of October in 2019, and from beginning of April to
end of October in 2020, 2021 and 2022). This precipitation reduction corresponds to the 1st quantile
of the local 100-year precipitation record (Swedish Meteorological and Hydrological Institute, 2021).
Each shelter covered two plots, one for the drought treatment and one for the combined drought-
compost treatment. A rubber sheet, approximately 40 cm in depth, was inserted in the soil around
each shelter to isolate the study plots from the ambient soil moisture. Pictures and sketches of the
sites and of the experimental design are presented in Roth et al. (2023).
**Soil and vegetation sampling and analyses**
Soil and root samples were collected in all plots at the end of the first growing season in 2019 (August-
September), and again at the end of the experiment in 2022 (August and October). Samples for soil
bulk density were collected to a depth of 45 cm with a large fixed volume root auger with a sharpened
cutting edge (8 cm diameter; Eijkelkamp, The Netherlands). The cores were taken incrementally
every 15 cm and then divided in 5 cm segments, and the bulk density was determined after drying the
samples at 105 °C. After drying, a subsample from the same core was used to calculate the soil organic
matter (SOM) content through loss on ignition at 550 °C. A subset was further burned at 960 °C in
order to determine the presence of inorganic C, which was very low (0.5 %), indicating that the total
C can be considered equal to organic C (OC). Samples for total C and N and $\delta^{13}$C were taken to a
depth of 1 m with a Pürckhauer soil corer (2.5 cm diameter; Eijkelkamp, The Netherlands). The
analyses for total C and N and $\delta^{13}$C were carried out by the Stable Isotope Facility at UC Davis
(California). A subset of these samples was sent to a commercial lab and used for pH measurements
(Mantech Automax 73, Guelph, ON., Canada) and nutrient content analyses (P, Ca, Mg and K; Avio
500 ICP Optical Emission Spectrometer, Perkin Elmer, Waltham, MA; USA). Soil moisture was
measured every three weeks throughout the growing season (2019 through 2022) from one access
tube (1 m long) permanently installed in each plot, using a PR2 profile probe (Delta-T Devices Ltd,
Cambridge, UK). The values used in the analyses are growing season averages of volumetric soil
water content (%) in the first 30 cm in each plot. Root biomass was collected in September 2019 and



in August 2022 with one soil core sampled with a root auger (8 cm diameter; Eijkelkamp, The
Netherlands) to a depth of 30 cm in all plots and to a depth of 45 cm in a subsample of 16 plots, with
soil cores divided into 5 cm segments. The roots were rinsed with water on a 0.5 mm mesh sieve to
remove soil and then scanned, followed by drying at 60 °C for 48 h to obtain the dry weight. The
scanned images were analyzed with WinRhizo (Regent Instruments, Québec, CA) to obtain root
volume, length and diameter, used to calculate root mass density ($g_{roots}$ $cm^{-3}_{soil}$), specific root length
($cm$ $g^{-1}_{roots}$) and root tissue density ($g_{roots}$ $cm^{-3}_{roots}$). Aboveground biomass was harvested from one
quarter (1 $m^2$) of each plot every year in mid-July, by cutting at ground level (including moss and
dead biomass). More details of the sampling design are presented in the Supplements (Table T1).
**Statistical analyses**
Total C content (as % of total soil mass) and soil C stocks normalized by soil sample thickness ($kg/m^3$)
were calculated for all 48 plots using the total C content data (available for a subset of the samples)
and the bulk density and SOM data (available for a complementary subset of the samples). A
regression was performed to calculate SOC from SOM data and thus obtain a complete dataset,
$SOC = 0.328 \times SOM + 0.217,$     (1)
where SOC and SOM are expressed in $kg/m^2$ (Fig. S1).
The fraction $F$ of compost-derived C remaining in the soil in year 2022 was calculated with a two
end-member mixing model, as in Poeplau et al. (2023),
$F = \frac{\delta^{13}C_{compost\,treatment} - \delta^{13}C_{control}}{\delta^{13}C_{compost} - \delta^{13}C_{control}},$     (2)
where $\delta^{13}C$ was measured in both compost-amended (compost or compost-drought) and control (no
compost or drought-no compost) plots.
All the results and statistical analyses are limited to the depth range of 0-45 cm. This is because this
soil depth contains the majority of the root biomass (95% within the first 30cm, mean ~17 cm) and
of the microbiological activity, and no effect of treatments could be detected below this range (data
not shown).
All analyses were made in R (version 3.3.3; R core Team 2017), and statistical models were designed
with the lmer function (package: lme4). Pairwise comparisons between categorical variables were
made with lsmeans (package: emmeans) and p-values were obtained with the ANOVA function and





the lmerTest package. Residuals from the models were checked graphically. Effect sizes were
obtained by calculating Cohen's d, with the formula
$$d = \frac{\overline{x_1} - \overline{x_2}}{s},$$ (3)
where $\bar{x}_1$ and $\bar{x}_2$ are mean values for the two groups for which the effect size is calculated, and S is
the standard deviation.
The effect of the treatments was tested on all plots from the 2022 dataset, including root biomass and
root traits, for which the values were log-transformed first. The model included compost (categorical
variable), drought (categorical variable) and sampling depth (continuous variable) as fixed factors
and plot (nested within site) as random factor. Cohen's d was calculated using the standard deviation
of the control group. The effect of the compost amendment on the C isotopic ratio was tested with a
model that included compost and depth as fixed factors, and plot (nested within site) as random factor.
Changes in soil C, bulk density and C stocks were also tested with a model using depth as categorical
variable, to assess if changes occurred at specific depths. The landscape variability was tested on all
data collected in 2019 and from the control plots in 2022. The model included grassland site, catenary
position and sampling depth (continuous variable) as fixed factors and year and plot as random
factors. Cohen's d was calculated using the standard deviation pooled from all groups. Temporal
changes during the experiment not caused by the treatments were tested using data obtained in 2019
and 2022 from the control plots. The model included year and sampling depth (continuous variable)
as fixed factors and plot (nested within site) as random factor. Cohen's d was calculated using the
standard deviation of the 2019 dataset. The variable depth was not included in the models for
aboveground biomass.

## Results

Drought decreased soil moisture by 16% in the upper 0-30 cm. The effect of drought was consistent
over sites, years and seasons, and there were no statistically significant differences in the drought-
driven soil moisture loss between locations, years, or between spring, summer or growing season.
There was also no significant difference in soil moisture decrease between drought plots and drought-
compost plots. Additionally, the compost addition did not have any significant effect on soil pH or
on soil P, Ca, Mg and K. The compost addition did, however, raise the value of $\delta^{13}$C in the treated
plots (mean control plots = -27.44‰, mean compost plots = -27.10‰, P < 0.01), and the difference



was significant at 0-5 cm, 30-35 cm and 40-45 cm. The mixing model (Eq. 2) indicated that after
three growing seasons, the percentage of compost-derived C was 3.43 % in the 0-5 cm layer, 4.88 %
in the 30-35 cm layer and 5.51 % in the 40-45 cm layer in the compost plots, and 4.55 % in the 0-5
cm layer, 6.52 % in the 30-35 cm layer and 2.96 % in the 40-45 cm layer of the compost x drought
plots.

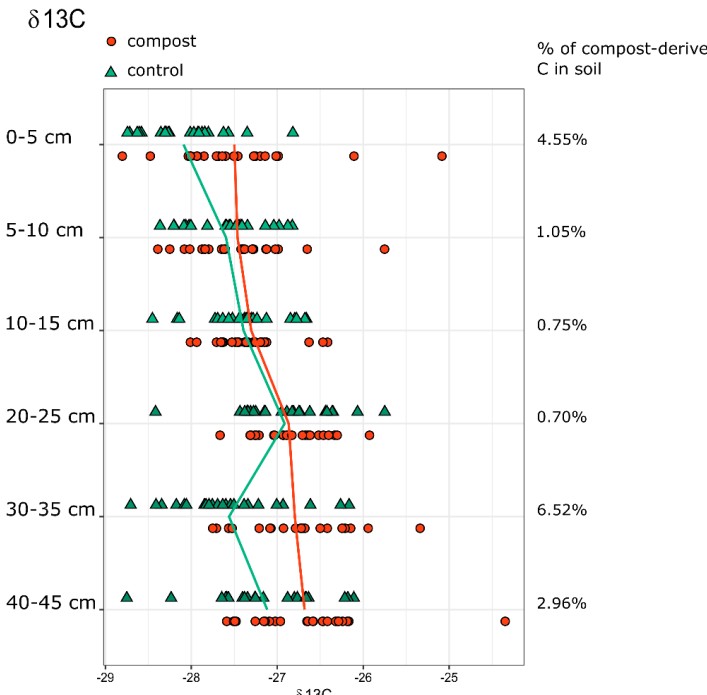


*Fig 1. Values of $\delta^{13}C$ in the soil in compost-treated (red dots) and untreated (control, green triangles) plots in 2022, at*
*different depths. The % of compost-derived C in the soil was calculated with a mixing model (Eq. 2).*

### Compost and drought effects

The compost treatment increased total soil C content (P = 0.04) and aboveground biomass (P < 0.01).
The latter increased by 23% (mean control plots = 642 g m$^2$ ± 129.23, mean compost plots = 788 g
m$^2$ ± 221.7). The effect on soil C was significant only in the topsoil (0-5 cm, Fig. 2), where the relative
increase of soil C content was 18% (mean control plots C content = 2.99% ± 1.03, mean compost
plots = 3.53% ± 0.75). Soil nitrogen (N) was higher in the topsoil in the compost-treated plots (mean
control plots = 0.24% ± 0.06, mean compost plots = 0.28% ± 0.06), but the treatment did not affect



the C:N ratio. The compost treatment also decreased bulk density by 9% (P = 0.03) in the first 10 cm
of soil (mean control plots = 1.34 g cm$^3$ ± 0.18, mean compost plots = 1.22 g cm$^3$ ± 0.17). The compost
did not have any statistically significant effect on other variables.
Drought only had an effect on aboveground biomass, which decreased by almost 4% under the rainout
shelters (mean control plots = 642 g m$^2$ ± 129.23, mean drought plots = 617 g m$^2$ ± 180.25). The
increase in the soil C content under compost addition was offset by the lower bulk density, so that
there was no statistically significant change to soil C stocks. However, we note that mean soil C
stocks were 6% higher in the compost-treated plots in the first 15 cm, slightly higher than the
percentage of compost-derived C found in that layer (mean control plots = 4.02 kg m$^2$ ± 0.92, mean
compost plots = 4.26 kg m$^2$ ± 0.59).



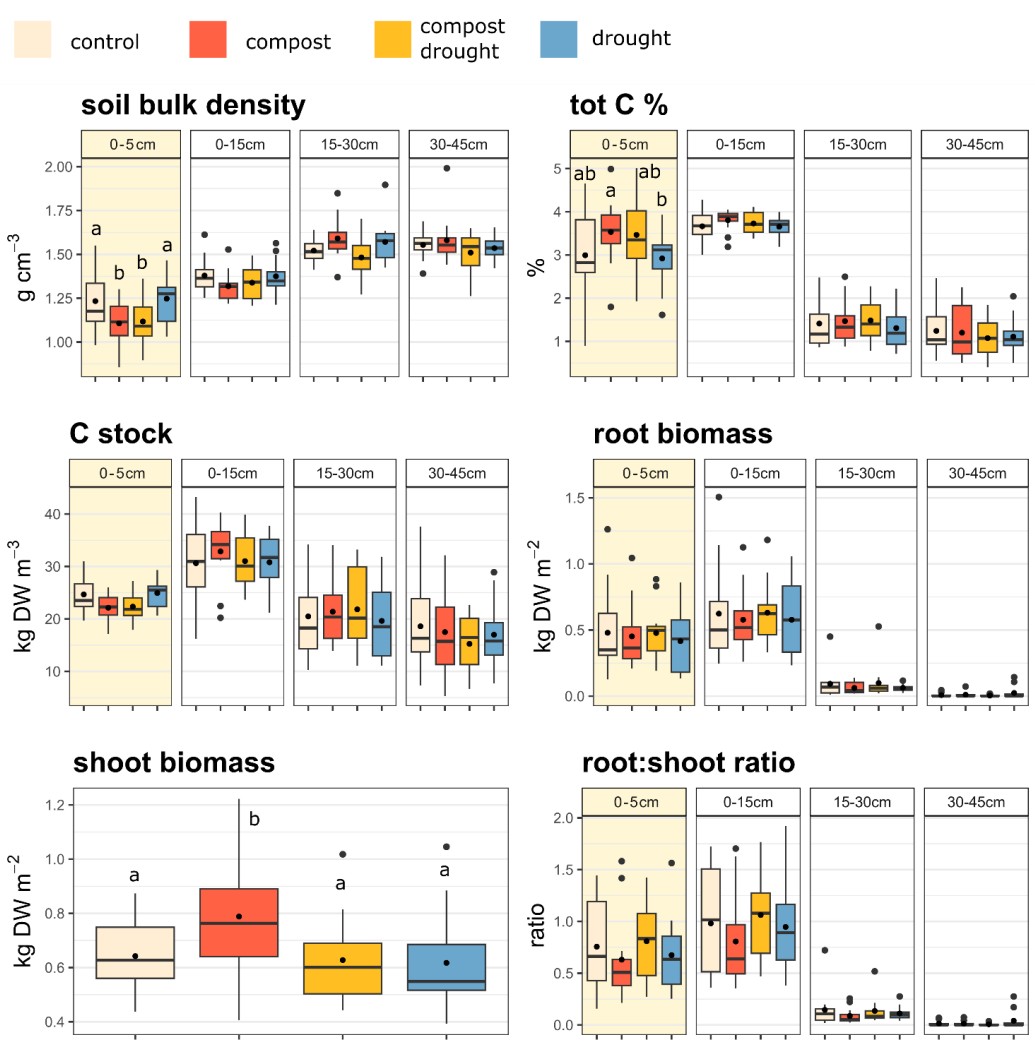

*Fig. 2. Values of soil bulk density, total soil C contents, soil C stocks, root biomass, shoot biomass and root-shoot ratio, at different sampling depths in 2022. White = control, red = compost, yellow = compost×drought, blue = drought. Bars show mean (dot inside the bar), median (horizontal line) and interquantile range (IQR, colored bar); whiskers extend to 1.5×IQR; dots in the graph are outliers. Different letters indicate statistically significant differences between means (P < 0.05).*

**Root traits**

Drought led to an increase in root tissue density (P = 0.048), in specific root length of fine roots (P = 0.049), and in average root diameter (P = 0.045). If only roots in the topsoil (0-5 cm) were considered,



in addition to the patterns above, specific root length of coarse roots decreased under drought (P =
0.04), while root tissue density (P = 0.02) and specific root length of all roots increased after compost
addition (P = 0.01).

In all control plots, soil C and root biomass was correlated both in the topsoil (5-10 cm, r = 0.42, P =
0.04; 10-15 cm, r = 0.5, P = 0.01) and in the 0-30 cm layer (0-30 cm, r = 0.63, P < 0.01). Soil C
content was also correlated to the root:shoot ratio (5-10 cm, r = 0.44, P = 0.03; 10-15 cm, r = 0.4, P
= 0.052; 0-30 cm, r = 0.43, P = 0.04). In the compost treated plots, the only significant correlation
was between soil C and root biomass when considering the whole 0-30 cm layer (0-30 cm, r = 0.55,
P < 0.01). The correlation between soil C and aboveground biomass remained constant in both control
and compost-treated plots (r = 0.22, P < 0.01 in both groups). This indicates that the compost
treatments affected soil C in the topsoil and aboveground biomass more than they affected roots and
deeper soil.

**Landscape spatial variability**
Soil C contents, total C stocks, bulk density, root biomass and root:shoot ratio all showed statistically
significant (P < 0.05) differences between catenary positions and depths, and soil C content and bulk
density also differed significantly between grasslands (Fig. 3, Table T3). Grassland identity and the
interaction between grasslands and catenary positions were the only significant predictors of
aboveground biomass, suggesting this variable is most likely related to land-use history and plant
community composition.





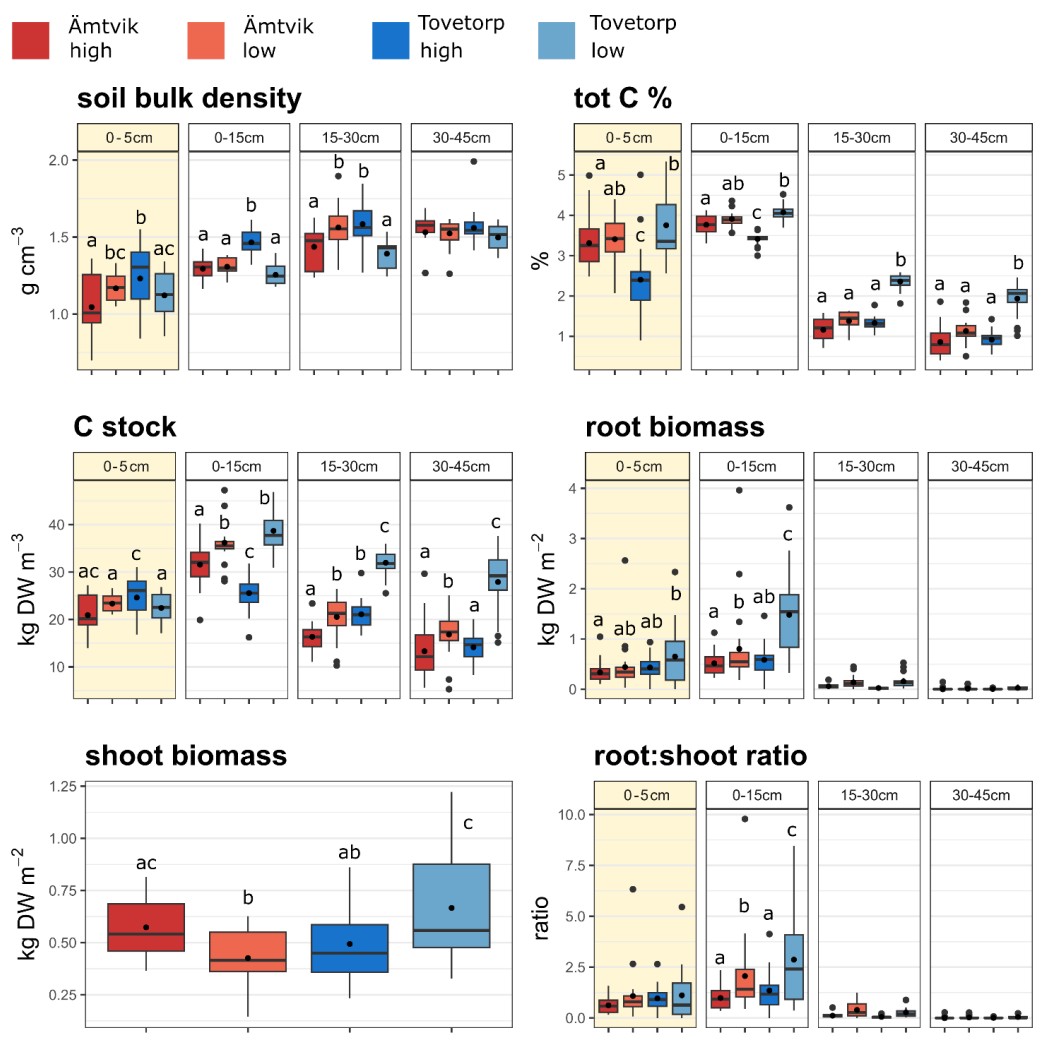


*Fig. 3. Values of soil bulk density, total soil C content, soil C stocks, root biomass, shoot biomass and root-shoot ratio,
at different sampling depths at the four sites, excluding treatments. The data consists of average values from 2019 (all
plots) and 2022 (only control plots). Red = Amtvik High, orange = Ämtvik Low, blue = Tovetorp High, light blue =
Tovetorp Low. Bars show mean (dot inside the bar), median (horizontal line) and interquantile range (IQR, colored bar);
whiskers extend to 1.5×IQR; dots in the graph are outliers. Different letters indicate statistically significant differences
between means (P < 0.05).*



**Natural changes during the 2019-2022 period**
From 2019 to 2022, we observed large changes in soil C and plant biomass in the control plots. Soil
C contents, total C stocks, bulk density, root biomass and root:shoot ratio all showed statistically
significant ($P < 0.05$) differences between 2019 and 2022 and between depths (Fig. 4, Table T4).
Aboveground biomass also differed significantly between sampling years. Between 2019 and 2022
total soil C contents and root biomass in the control plots decreased by 10.7% (from 3.35% ± 1.05 to
2.99% ± 1.03) and 8.4% (from 522.96 g m$^{-2}$ ± 626.48 to 479.25 g m$^{-2}$ ± 320.75), respectively, in the
first 5 cm, and by 27.1% (from 2.77% ± 0.78 to 2.02% ± 0.61) and 67.4% (from 477.26 g m$^{-2}$ ±
252.49 to 155.81 g m$^{-2}$ ± 51.62), respectively, in the first 15 cm. Aboveground biomass instead
increased by 53% (from 419.68 g m$^{-2}$ ± 137.45 to 642.23 g m$^{-2}$ ± 129.23).



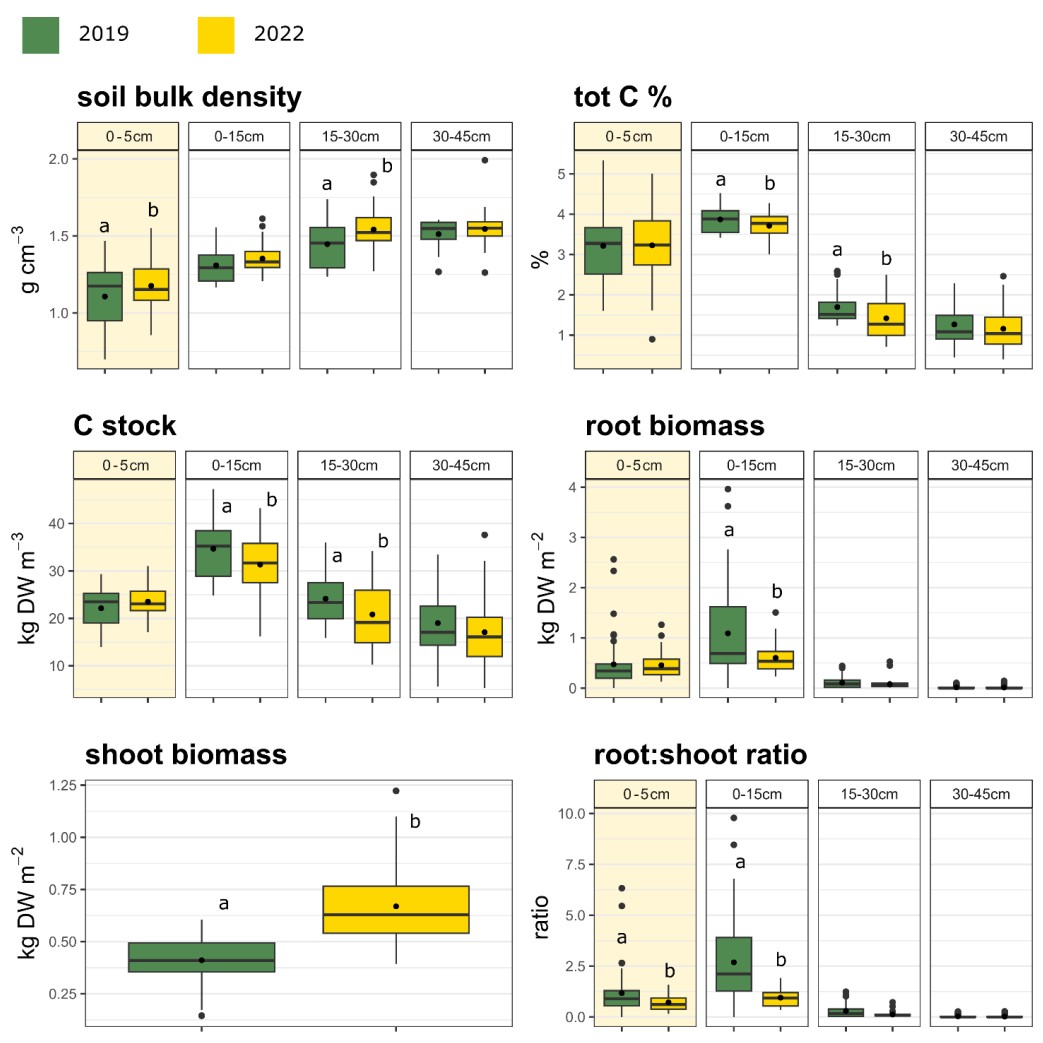


*Fig. 4. Values of soil bulk density, total soil C contents, soil C stocks, root biomass, shoot biomass and root-shoot ratio, at different sampling depths in 2019 and 2022 (excluding treatment plots). Values are means for all plots. Green = 2019, yellow = 2022. Bars show mean (dot inside the bar), median (horizontal line) and interquantile range (IQR, colored bar); whiskers extend to 1.5×IQR; dots in the graph are outliers. Different letters indicate statistically significant differences between means (P < 0.05).*



Table 1. Effect sizes (Cohen's d) of the differences between sites, years and treatments for each soil depth. Effect sizes can be regarded as small (absolute value <0.2, black), medium (0.2-0.8, orange) or large (> 0.8, red).

| depth | factor | Tot C | C stock | Bulk density | Root biomass | Aboveground biomass |
|-------|--------|-------|---------|--------------|--------------|---------------------|
| 0–5 cm | grassland | 0.21 | 0.10 | -0.40 | -0.32 | |
| 0–15 cm | | 0.64 | 0.58 | -0.36 | -0.24 | -0.17 |
| 15–30 cm | | -0.67 | -0.89 | -0.01 | 1.00 | |
| 30–45 cm | | -0.53 | -0.51 | 0.53 | 0.28 | |
| 0–5 cm | catenary position | -1.03 | -1.12 | -0.06 | -0.44 | |
| 0–15 cm | | -0.61 | -0.47 | 1.09 | -0.86 | 0.14 |
| 15–30 cm | | -0.48 | -0.46 | -0.68 | -0.64 | |
| 30–45 cm | | -0.96 | -0.99 | 0.62 | 0.27 | |
| 0–5 cm | year | 0.34 | -0.08 | -0.81 | 0.07 | |
| 0–15 cm | | 0.96 | 1.25 | 0.06 | 1.27 | -1.62 |
| 15–30 cm | | 0.21 | 0.23 | -0.19 | 0.49 | |
| 30–45 cm | | 0.30 | 0.13 | -1.41 | 0.87 | |
| 0–5 cm | compost | 0.52 | 0.26 | -0.79 | -0.08 | |
| 0–15 cm | | -0.02 | 0.26 | 0.40 | 0.14 | 1.13 |
| 15–30 cm | | -0.13 | -0.18 | 0.50 | -0.33 | |
| 30–45 cm | | -0.49 | -0.33 | 1.25 | 1.82 | |
| 0–5 cm | drought | -0.07 | 0.01 | 0.09 | -0.20 | |
| 0–15 cm | | -0.29 | -0.22 | -0.12 | 0.95 | -0.19 |
| 15–30 cm | | -0.33 | 0.09 | 0.93 | -0.49 | |
| 30–45 cm | | -0.39 | -0.63 | -1.03 | 0.24 | |

# Discussion

## Compost effects on soil C and plant growth

Total soil C contents increased after compost application, but because bulk density was also reduced, there was no significant increase in soil C stocks (confirming our first hypothesis), despite higher



mean values of soil C per m$^2$ in the compost treated plots in the first 15 cm of soil. This difference was lower than the estimated C addition (~0.54 kg C m$^{-2}$), likely due to respiration loss. Compost can be considered a recalcitrant type of organic amendment, with initially slow but persistent effects expected to be observed years after the first application (Sarker et al. 2022). Therefore, it is unlikely that an effect of the treatment on soil properties and soil C had occurred before our 2022 sampling, and that such an effect was somewhat transient and undetectable at the time of the sampling. This conclusion is also supported by the isotope tracing, indicating that at least a fraction of the compost-derived C is still present in the soil after three growing seasons. In addition, the significant increase in aboveground biomass three years after the compost application indicate the persistence of favorable plant growing conditions, such as increased N in the soil. These results are in accordance with Fenster et al. (2023), who found that the benefits of compost treatments on the ecosystem C balance of grasslands one year after application were manifested as extended growing season, and thus potentially higher plant productivity, rather than as an increase in net soil C. This also stresses the importance of including vegetation dynamics when assessing the effectiveness of C management.

Compost enhanced aboveground biomass growth, but not root growth, thereby only partly confirming our first hypothesis, and suggesting the presence of a tradeoff between root and shoot investment. This was already observed in Garbowski et al. (2020) and is in line with the expectation that plants in nutrient-rich environments can allocate to aboveground tissue growth the resources that would otherwise be allocated to nutrient acquisition belowground (Bloom et al. 1985; Poorter and Nagel 2000). In broader terms, this suggests that the compost treatment shifted the C balance between soil pool and vegetation pool, and moved the plant C allocation from belowground- to aboveground organs. Nevertheless, increased root tissue density and specific root length in the topsoil suggest that root response to organic amendments is manifested in more subtle changes in root traits, rather than in net root biomass production.

Our experimental setup did not allow us to test whether microbial activity and microbial biomass increased as a result of compost addition, as was reported by previous studies (Sarker et al. 2022; Gravuer et al. 2019). However, the limited effects of the compost treatment on soil C stocks suggest that the C sequestration benefits in the form of increased plant growth might have been offset by increased microbial respiration (promoted by either compost or enhanced rhizodeposition of more



productive plants). Finally, the significant spatial and temporal variability in both soil C and
vegetation biomass observed in the control dataset suggests that treatment effects might be site-
specific (Garbowsi et al. 2020), and management plans seeking to optimize soil C sequestration
should consider the potentially interactive effects of several biotic and abiotic factors. For instance,
the increase in aboveground plant biomass after compost application was mostly driven by the grass-
rich plots in the Tovetorp grassland (Roth, 2023), suggesting that plant community composition might
be important in determining the effects of soil amendments on grasslands.

**Drought effects on soil moisture, soil C and plant growth**

Drought treatments reduced soil moisture and aboveground plant biomass but did not significantly
decrease root biomass (Table T2), indicating a tradeoff between above- and belowground biomass
investment. Because plant growth is very sensitive to yearly fluctuations and even intra-annual
distribution of precipitation (Knapp and Smith 2001, Porporato et al. 2006), and because our analyses
are based on only two temporal datapoints (2019 and 2022), it is difficult to assess whether drought
reduced plant turnover, defined as the ratio of standing biomass to net primary productivity (NPP).
We note that while the precipitation in the growing seasons 2019 and 2022 (April through August)
was roughly the same (157 mm and 156 mm, respectively), the 2019 sampling followed an extremely
dry summer in 2018, when the study area received only 77 mm of precipitation, about half of the
precipitation compared to the average 1961-1990 (historical data from SMHI, 2021). Conversely, the
2022 sampling followed the very wet 2021, when the area received almost 140% of the normal
precipitation over the same time period (250 mm). It is possible that a legacy effect of these two
precipitation extremes may have affected plant growth, particularly aboveground (Fig. 4), where
growth is more sensitive than root biomass to yearly fluctuations in water availability (Zhang et al.
2021). Legacy effects of the 2018 drought could have hampered growth in 2019, as aboveground
vegetation in the control plots increased by more than 50% between 2019 and 2022. Conversely, the
high summer precipitation in 2021 could have buffered the effects of the experimental drought in
2022, leading to overall weak drought effects.
The drought treatment had a relatively small impact on plant biomass (Fig. 2). In addition to potential
effects of interannual precipitation variation, this may be due to adaptation in the plant community
during the treatment years (Basu et al. 2016), or that the drought was not intense enough. Roots in



particular where not significantly affected by drought (Fig. 2), but while we monitored the relative
proportions of annuals and perennials, grasses and forbs in each plot, we do not know which plant
species the sampled roots belong to. Therefore, we cannot make any conclusions related to the
ecology of these plant groups, all of which can be expected to respond differently to drought (Zhang
et al. 2017; Mackie et al. 2019; Zhong et al. 2019). However, since the magnitude of the drought did
not differ between locations and since soil physical properties were similar across sites, we can
hypothesize that differences in the plant communities account for at least some of the spatial
heterogeneity observed in our study, as was observed in Garbowski et al. (2020). Also, while drought
effects on root biomass were marginal, the drought treatment did increase both root tissue density and
average root diameter, suggesting adaptation of root traits in these plant communities.
Adopting a standardized approach for the drought experimental design makes our findings easier to
compare with others, but partial rainout shelters will still allow for a substantial amount of
precipitation to pass through the roof sheets. If there is enough precipitation, even the small
percentage of rain that reaches the ground might bring the soil moisture over the threshold of the
permanent wilting point. It is also possible that soil water retained in the soil from snowmelt or
winter/spring precipitation could have sustained vegetation growth in the drought treatments. Finally,
experimental droughts do not control for reduced air humidity, which may underestimate negative
responses of plant biomass to drought in field experiments (Kröel-Dulay et al. 2022), and for
increased temperatures, which often occur in combination with natural droughts. Drier and warmer
air increases evaporative demand, causing stomatal closure and thus lower productivity for a given
soil moisture level (Zhang et al. 2019).
To understand the ecosystem-level implications of drought, soil C changes need to be considered as
well. Dry conditions decrease heterotrophic respiration because microbial activity is inhibited due to
both physiological mechanisms, such as osmoregulation diverting efforts from resource acquisition
to survival, and physical mechanisms, like the slower transport of substrates in dry soils (as the water
films around soil particles shrink and pore connectivity is lost) (Moyano et al. 2013; Schimel 2018).
However, heterotrophic respiration increases again after soil rewetting, leading to disproportionally
large C emissions during the short post-rewetting period (Canarini et al. 2017; Barnard et al. 2020).
In our experiment, drought had no effects on soil C contents and stocks, as per our second hypothesis,



but it slightly reduced soil bulk density (in a pre-treatment vs post-treatment comparison, data not shown), possibly in relation to shrinkage in dry soil. Because drought reduced plant productivity (and thus C inputs to soil), the lack of drought effect on soil C stocks can be explained by a reduction of microbial activity approximately of the same magnitude as the reduction in plant productivity. This is supported by the fact that in the topsoil, the drought plots with added compost had a higher fraction of compost-labelled isotopes compared to the non-drought plots. Therefore, any soil C emission pulses at rewetting were not sufficient to compensate for the lowered microbial activity during the soil moisture dry-downs.

**Interactive effects of compost and drought**

The effects of soil amendments on water retention capacity are modulated by soil texture, by the quantity and quality of soil organic matter (Rawls et al. 2003; Yang et al. 2014; Franco-Andreu et al. 2017; Sarker et al. 2022) and by the chemical composition of the compost (Franco-Andreu et al. 2017). In our study, soil moisture did not differ between the drought plots and the ones with drought and compost, which indicates that the soil amendment did not increase soil moisture, in contrast with previous findings (Franco-Andreu et al. 2017; Ali et al. 2017), and leads us to reject our third hypothesis. Interestingly however, while both compost and drought slightly reduced root biomass, the compost applied on the drought plots led to an increase in root:shoot ratio (Fig. 4). While plants may reduce their belowground biomass investment when adding organic matter, this mechanism appears to work differently under drought, and the observed shift in C allocation belowground may serve to aid in water acquisition. This, in turn, could lead to increased evapotranspiration, and this improved capacity for soil water absorption could mask any increase in soil water retention capacity in the compost x drought plots. However, since our experiment did not include drought recovery, it is not known if this change would persist after the end of the experimental drought.

## Conclusions

The goal of this study was to provide an overview of the changes in soil, above- and belowground vegetation C content and biomass within a grassland ecosystem, through a multifactorial drought and compost amendment field trial. The compost treatment revealed contrasting responses of shoots and roots, but it did ultimately not result in an increase in soil C stocks. Drought decreased aboveground biomass, but root response was limited to shifts in root traits. Compost amendment and drought had



distinct effects on plant C allocation, revealing the presence of trade-offs in their responses to environmental change. These findings improve our understanding of C dynamics in grasslands, offering potential contributions to ecosystem C modelling. We also observed significant spatial and temporal variability in vegetation and soil C dynamics over the study period, which may be driven by differences in topography, land use and plant community composition. This suggests that ecosystem C dynamics can be influenced by multiple biotic and abiotic factors, which can be revealed by field observations and multifactorial experiments.

## Author contributions

Conceptualization and methodology: SC, DG, GH, SM, NR; Field investigation and lab work: DG, NR; Statistical analysis: DG; Writing – original draft: DG; Writing – review & editing: DG, SC, GH, SM, NR.

## Acknowledgements

The authors wish to thank Tovetorp Research Station and the land owners of Ämtvik for allowing us to work on their grasslands, Linnea Ström, Louis Hunninck, Luca Durstevitz, Marie Förg, Olov Hammarström, Victor Eriksson and Willeke A'Campo for help in the sampling, and Lukas Rimondini, Torge Gerwin and Sean Burke for help in the lab. Funding for this study was provided by the Bolin Centre for Climate Research and by the Mannerfelt Fund.

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
