# Peer review of "Experimental drought and soil amendments affect grassland above- and belowground vegetation but not soil carbon stocks"

_EGUsphere, 2023_

## Referee Comment (RC1)

**General comments:**

The manuscript needs to be sharpened in all parts.

The results shown differ from what is described in the discussion, the interpretation of the results is often about why the data did not show certain expected patterns when it should be about what their data actually did show. In the current form, I would not recommend publication. The manuscript needs a bit more than major revisions and my suggestions are below.

Scientific significance: 3-4

Scientific quality: 2

Presentation quality: 3

1. **Does the paper address relevant scientific questions within the scope of SOIL?**

   Yes it does. It addresses drought x compost effects on plant growth and parameters relevant for evaluation of soil carbon development.

2. **Does the paper present novel concepts, ideas, tools, or data?**

   It presents new data on drougt x compost effects in grasslands

3. **Does the paper address soils within a multidisciplinary context?**

   Yes.

4. **Is the paper of broad international interest?**

   Generally, the topic itself is of international interest.

5. **Are clear objectives and/or hypotheses put forward?**

   Yes.

6. **Are the scientific methods valid and clear outlined to be reproduced?**

   Yes.

7. **Is the soil type/classification adequately described?**

   No, not yet.

8. **Are analyses and assumptions valid?**

   Yes.

9. **Are the presented results sufficient to support the interpretations and associated discussion?**

   Not always, especially the interpretations on the microbial community and it was not explained how plant community structure was assessed. Often, the results shown and the discussion/interpretation differs. E.g. in the discussion they say there was shift of C allocation in plants towards roots while a) not measuring C content of the roots and b) no change in root biomass is seen in the results. This happened regularly, especially in the conclusions.

10. **Is the discussion relevant and backed up?**

Yes and no. There are relevant parts, there are parts that can be cut and there are parts that do not match the data (as described above)

11. **Are accurate conclusions reached based on the presented results and discussion?**

No. The conclusions and parts of the discussion do not match the results.

12. **Do the authors give proper credit to related and relevant work and clearly indicate their own original contribution?**

Yes.

13. **Does the title clearly reflect the contents of the paper and is it informative?**

No. The reason is named below.

14. **Does the abstract provide a concise and complete summary, including quantitative results?**

No, not yet but can be achieved.

15. **Is the overall presentation well structured?**

Yes.

16. **Is the paper written concisely and to the point?**

Can be improved, see my comments below.

17. **Is the language fluent, precise, and grammatically correct?**

Mostly yes.

18. **Are the figures and tables useful and all necessary?**

Yes. Sometimes, a figure mentioned in the text is wrongly numbered.

19. **Are mathematical formulae, symbols, abbreviations, and units correctly defined and used according to the author guidelines?**

Yes.

20. **Should any parts of the paper (text, formulae, figures, tables) be clarified, reduced, combined, or eliminated?**

Yes, see my comments below.

21. **Are the number and quality of references appropriate?**

Not yet, this can be improved.

22. **Is the amount and quality of supplementary material appropriate and of added value?**

Yes, but in the methods part, it is not described how macronutrients e.g. Mg or P have been measured thus, there is value lost.

**Major comments:**

The title is misleading. The main topic of the text is about compost and drought effects. The spatial and temporal variability in soil and vegetation is only mentioned a few times and the hypothesis also focus on the compost and drought effects.

**Abstract:**

L9: grassland soils have no large C sequestration potential, only a change of management from arable land to grassland has this C sequestration potential. Grassland itself not. Grassland it self is either in a C equilibrium if it is a grassland for long enough or C accrual may happen which may sequester C depending on how this accrual is achieved. Only a global net increase in C is sequestration, otherwise if e.g. compost is used, the C contained in this composed is merely moved from site 1 to site 2 and does not achieve a net uptake of C from the atmosphere into the soil.

L14: Vegetation C pools are not shown. Biomass is shown.

L14-15: Too fuzzy. You could say that in the compost addition treatment you found higher aboveground biomass. There is a correlation, but you did not prove the causation. Also increased compared to what? The control? Mention this to be clear.

L16: decreased it only compared to compost treatment. Compared to the control nothing happened.

L17-18. Not true based on your data. Soil amendment shifted C allocation upwards as shoot biomass increased, yes. But drought had no effect on belowground biomass. Only the type of roots was affected, not the biomass. Thus, drought had no C shift as a consequence.

L21: Nowhere, in the complete manuscript do you talk about upscaling. This part needs to be cut

**Introduction:**

L31: "…mitigate soil organic carbon loss…" I would add here that this is the case locally. E.g. mitigate soil organic carbon loss on a specific site. This is relevant, as leakage needs to be taken into account as all amendments are basically biomass from another site, transformed and then brought to the specific site where you want to amend this biomass. You export biomass from site A, reducing C inputs there, and import it to site B, increasing C there. Also, leakage should then be explained to the readers, so they can follow. This matters as this is the major limitation to such amendments. It needs to be additional biomass globally, not locally, to enable C sequestration in soils by increasing SOC accrual.

L86: The text before does not highlight enough the time scales of SOC formation. Sometimes you refer to land use history which indirectly shows that the effects are decadal up to centennial and more. However, then you conduct three year experiment to search for SOC stock changes which is a very short time for C stabilization experiments. This needs to be mentioned in the discussion the latest. Of course, you find an increase in SOC, as you add C rich material. This is not surprising. But how much of this would be left after 10 years, especially of a one time only amendment which is also not the agricultural reality. This is a drawback of the study that needs to be discussed in the discussion part which it currently is not.

L81-89: This belongs partially in the methods and in the results part and not in the introduction.

L23-90: The introduction in general, needs to be more focused. You start with general topics, good. Then you need to narrow everything down to the question: Why do we need to study organic

amendments more? Show clearly the knowledge gaps and show ways to answer these. These ways will be your hypothesis. This link needs to be clearer (I address this in the specific comments section). In the end, when the readers read your introduction, they need to be able to not only follow your content but also to reach the same conclusion about why does your topic matter and how to fill the knowledge gaps. As of now this is too weak and not focussed enough. When I read your hypothesis, I wonder why you look at organic amendments and not mineral amendments or other management practices. It still seems a little bit "random". With a little bit of work, this can be changed and improve the overall quality of the paper.

L91-94: Hypothesis 1: How does compost increase plant growth? Make the mechanism clearer in the introduction (line 58 needs more information on this for the readers to follow). Also, how does compost increase soil C? You did not state that compost is an organic amendment and you did not state that organic amendments are having high C content. Needs to be added in the introduction. L37 states this indirectly which is too fuzzy. Make this connection clearer by explaining e.g. how compost is made and its properties. Then this part of the hypothesis becomes very easy to follow. It remains also too fuzzy how a decrease in bulk density causes a negative effect on C stocks. Everything up to here is known and not suitable for a hypothesis, the real hypothesis follows now "we expect that these mechanisms have counteracting effects on net soil C storage;" This hypothesis is not clear and too fuzzy. Which is the direction of the effect you expect based on the introduction part? Do you expect the positive effect to be stronger or the negative one? This would make for a great hypothesis.

L94-95: I recommend to turn this around to be clearer in your communication: Drought will have a weak or non-detectable effect on SOC by decreasing both productivity (organic C input) and respiration (microbial decomposition of SOM). In addition: as for hypothesis 1 it is not entirely clear why you study drought effects. You need to stress this more in the introduction as you address there rather the need for SOC increases to improve soil health and that this could happen via amendments. The climate effect is only mentioned in 1 line or so. Stress this point, as it is highly relevant! Furthermore, be more precise. Will drought have an effect or not? If yes, in which direction do you expect it based on what should be contained in the introduction?

**Methods:**
L132-135: Too fuzzy. What the auger 15cm long and you took the 1$^{st}$ sample 0-15cm. Then you used the same hole for the next 15cm (15-30)and then the same hole for 30-45cm? Please specify how you did it. Also, in case you used the same hole and the auger was 15cm (and not 45cm) long: Do you not think, that the insertion of the 15cm long auger compacts the soil below 15cm and thus you would get much denser samples for 15-30 and 30-45cm?

L137-138: In what kind of increments or is this a bulk sample from 0-100cm? Please add.

L140: Was this measured pH the actual or the potential pH? Was measured in destilled water or CaCl2? Please add.

L147: How did you place the auger? On the plants, in between plants? This will be changing the results strongly, so please be more specific on how you did this.

L148: Why do you sample all plots to 30 cm and then just a subset of 16plots to 45cm? Which were the 16 plots? Which plots and why were they not sampled to 45cm? Please be clearer on this.

L157: How did you calculate the stocks? Explain and state which factors you included. Did you consider the packing density? For this, see publications of Poeplau et al, which you already cited for the d13C calculations. He explains how to calculate stock correctly.

L168-170: What do you mean by "This is because this soil depth contains the majority of […] the microbiological activity […]" What is a majority of activity? How did you measure this? You have not explained anything about microbiological measurements so I wonder how you could back you sentence up by data. Please adjust.

L174: what was you alpha? 0.05, 0.01? Please add.

**Results:**

L211: Figure 1 does not include the effect of the drought treatments. It would be nice to see them too in a similar figure, especially, as the text above L205-209 does not read well and takes some time to see in which treatment the C derived from compost is higher.

L232: Figure 2: Why is the 0-5cm box always yellow? Not explained. If there is no reason, I'd suggest to remove this background colour. Problematic is also that there is a dot in the box-plots for the mean value but the points outside the box-plots are outliers. If possible, adjust this to two different symbols.

The caption needs to be adjusted: there are no bars, these are called box plots or Box-Whisker-Plots. Please adjust this. Also explain in the caption what C is. Figures must be self-explanatory without the main text. So, the reader needs to learn again that C = carbon

L263: Figure 3 similar to above. The yellow background, the C=carbon, the inside and outside dots. In addition, the sampling depths need to be named in the caption. What does "low" and "high" in the sites name mean? Too fuzzy, I assume it is the upslope and downslope area? In the text you usually call higher position in the catena (or similar), therefore, keep it consistent and call it here the same.

L280: Figure 4: comments as above. In addition: Be more specific as you state (excluding treatment plots). Better write Values of X and Y …..in 2022 of all control plots. Also give us an n= ? How many control plots are meant? This should be a standalone figure. This n in addition, should be also added for all figures above. This helps the reader to understand the values better as it matters if those values are derived from 3 plots or from 100.

L185 Tabe1: orange and red are hard to distinguish. Also, people with a red-green weakness could not see any difference in colour. Depth with a capital D, Factor with F to keep the table headings consistent. Differences: Add the p value for that. This needs to be a stand alone table. Where do I see the sites names? In the figure before the sites had names, here not. Keep it consistent. Why are the horizontal lines around "year" thicker than all others? If I read catenary position as factor, by just looking at this table and nothing else, I do not know which positions you had, same for all other factors.  Name these.

**Discussion:**

The whole discussion needs to be checked if all statements about the results mentioned here, are actually found in the results. Comments on this will follow here.

L297: The effect of compost after the few applications is indeed expected to be hardly detectable. Here you could add results from long-term experiments showing after what time a significant change is to be expected.

L301-302: This conclusion is not entirely correct. It may be part of the reason, surely the climate of the growing season differed as well to have an effect. For the suspected increase in N (which you did

not measure) I suggest to add a reference showing this. In these lines be a bit more careful, as compost is one of multiple reasons why the biomass is be increased as you cannot prove that it was the compost treatment alone.

L302-305. Cut into two sentences to improve understanding. Also, what do you mean that compost application "was manifested as extended growing season"? I cannot follow the interpretations here.

L321: This is no C sequestration. This is C accrual. C sequestration in soils is always a global net increase. Compost is biomass exported from another area of land and imported on this plot. That entails a C loss in site 1 and an increase in site 2. Thus, just a shift of global stocks, not an increase. This is important as currently the discussion about the possibility of agriculture to offset GHG emissions is heated and lots of impossible hopes are there. Therefore, I strongly suggest in order to keep the communication clear, do not call compost amendments a C sequestration measure or similar. This is merely a management option to shift C from one site to another to improve soil health. C sequestration however is about offsetting climate change effects. In addition, as you state that there are no increases in soil C, even the term C accrual does not fit well either. Thus, better use potential C accrual.

L325: Also here, be careful with the term C sequestration. Here, I suggest to use …seeking to increase C accrual (or C stocks)….

L327: How can the Tovetorp grassland drive the increase in aboveground plant biomass? Unclear and fuzzy. What is a grass rich plot? Unclear. How can you make a statement about effects of plant community composition when you never measured it? The fact that C in the 0-5 cm plot was higher in that site could also have an effect, as you basically want to state more Soil C = more biomass. This whole conclusion here is not convincing yet and needs to be revised and sharpened. What is it that you want the reader to take as a take home message from this chapter? I could not tell yet. The last sentence here could for example be a statement/conclusion in 1 sentence about the compost effect on soil C (stocks?) and plant growth based on what you said before in this chapter. This (as of now lacking) key message would also allow the reader to better follow then the drought effects chapter.

L333-336: This statement in these lines is correct. Now take this into account when discussing my comment above for L301-302.

L338: Does your weather data availability allow to also look at the distribution of rainfall along the seasons? As you mention it in L333-334. That would be helpful.

L333-348: This paragraph could move to materials and methods as the site description. Here, you do not discuss your drought treatment effect and thus this section does not really fit in here.

L347-348: This is purely hypothetical and you have no reference nor data on this. I suggest to add references with a statement about how much this could in similar situations affect the data you have. If I only read the text as is, I could think that the result of the experiment is pure coincidence and that we could learn nothing from this data (which is not true!). Thus, put it into context and add a number about how much e.g the drought effects could be weakened.

L349-351: How so? I cannot follow how plant community fits in here and how this is linked to your results, as you did not measure plant community structure

L351-353: Where are those results?

L362-372: And how is this linked to your study and results? Be precise, do not imply it, state it and name the link. Also, this whole paragraph could be shortened and condensed.

L373-379: How does this link to your study and why is it relevant? The second half from L379 on is relevant, this part could be cut.

L379-388: You mentioned already before, that your experimental drought may not be intense enough, and you focus again on microbial activity. This whole chapter reads not like you explaining what you found/measured/observed but rather like: We did not find anything and here are the "excuses". This chapter needs rephrasing. Focus less (but mention!) what may have gone wrong (e.g. drought intensity) and then talk about what you found and what your data allows for interpretation. In addition, C storage in soils is a process that takes longer time scales than the time interval you measured. E.g. if you change your agricultural management (i.e. introducing cover crops to arable land) the C stocks may increase (through increased biomass and thus C input). This increase is not measureable after 3 or 4 years and not with so few samples as you have. E.g. for soil C monitorings timesteps of 10 years and thousands of samples are needed to identify reliably if C stocks actually did increase. This is a longterm process and thus I am not surprised that you did not find many differences. Apart from adding C via manure, which of course immediately increases the C stocks.

L396: What do you mean "slightly"? Significantly? If not, then better state "tended to reduce"

L397: Figure 4 is not showing this. There you see the treatments excluded. In Figure 2 you see the treatments. There is no difference in R:S ratio. This needs to be corrected and changed.

L397-403: Here, we would need references, if any of this will remain after the reanalysis of the comment of line 397.

L406: vegetation C content was one of the goals? Then why was there no data on the C content of the plants? Cut this part or rephrase.

L 407-408 No it did not. The changes in roots are insignificant and there is not even a pattern, for shoots you are right.

L408-409: No, it did not. Fig 2 shows that shoot biomass in the compost treatment was different from all other treatments.

L410: Where do you see these changes in C allovation? Not in Figure 2.

L411: How do they improve our understanding? Tell us.

L412: How can your findings contribute to improve modelling? Unclear and not discussed in the discussion part at all. Thus, this cannot be concluded.

L414: Why do you leave out the part about the effect of precipitation which you mentioned in the discussion?

L415-416: Yes of course. But this has been known before and is nothing new found by your study. This sentence could have been stated like this even without your study and thus does show the value of your study.

**Specific comments:**

L23-24: add some sources e.g. 4permille initiative, EU green deal, Farm to fork. Anything to show this increased interest

L27: "Sometimes": This makes me wonder, what it is called other times. Either cut the sometimes or give 1 or 2 more other names

L33-36: This is basically, what C sequestration in soils means. I would move this up and use the term C sequestration in soils thereafter.

L 34-43: You are right with what you say and here we need some literature references to back the mentioned topics up.

L37: "directly increases the standing stock of SOC" instead of "standing" I suggest to use "local" or "SOC stock of the site"

L42: Wording. The term "Soil C sequestration" is currently under debate as it suggests soil carbon is sequestered when in fact atmospheric carbon is being sequestered in soils. Thus, I suggest to write "potential C sequestration in soils"

L43: ….contributions *to soil C stocks*.

L43-44: You could also mention that roots are 2/3 more recalcitrant to decomposition and thus of a different quality that above ground biomass and thus better suited when trying to sequester C in soil

L44-47: Please add a reference or two

L52: "whether through conventional or regenerative methods" I would cut this part. It does not add any information and if you mention it you should clarify what "regenerative" methods are and what not. To avoid getting side tracked, I suggest to cut this.

L52-54: Already mentioned e.g. in L25-26. Redundant. Rather make a bridge from the last paragraph over to amendments.

L63: see my specific comment on line 42

L62-68: but this is also true for cropland. Why only consider grassland here when you talked about all agricultural land before?

L70: here you go back to crop yields e.g. cropland. Looking at the comment above, you switch randomly between all agricultural land, then specifically crop or grassland and back to all agricultural land. I suggest to harmonize it and talk about all agricultural land as long as possible until you get really into the details of grasslands and the topic you want to focus on.

L80: Agreed. But why on grassland? Stress the lack of data there more.

L86: Their? What does this refer to? The yearly droughts or the yearly droughts and the compost application? Fuzzy, please specify

L87: soil organic C measured and soil C stocks. Please explain to the reader how SOC and C differ from one another and how these are linked. If I know little about soil C I would wonder how you measure SOC and then derive C stocks and not SOC stocks. Too fuzzy and needs to be cleared up to ensure that everyone understands what you did.

L90: Why did you hypothesize this? This needs to be better linked to the introduction text.

L99-102: Move to M&M

L108: wording…I suggest: Today, the land management consists of grazing and hay production.

L112: It would be great to know the soil type for the 4 sites, e.g. are we talking about Cambisol and a more exact texture would be great, if available.

L115: Mention the value so we can see the difference.

L157-159: Consider to split this sentence into two. I needed to read it 3 times to follow along.

L167: "compost or drought-no compost)" add here a "respectively" after between "compost" and ")".

L184: "the C isotopic ratio" better delta 13C ratio. Be consistent.

L185ff: What is this "model"?

L187: "Landscape variability" Do you mean the position in the slope? Please explain.

L198: I suggest "The drought treatment" to be 100% clear, that this is a treatment and not a "normal" drought. In case the reader skips right to this chapter.

L200: You mention spring and summer (=seasons) then you mention also growing season. How is this defined, what time is the growing season? Please add.

L205-209: It would be informative to see the standard deviation of these numbers as well.

L214: This sentence is already an interpretation which belongs to the discussion part. I suggest to rephrase e.g. In the compost treatment total soil C content and aboveground biomass was increased.

L218: Was N significantly increased or not? Please add a p value

L220: "the C:N ratio" add: significantly. Because N content increases as you stated in the sentence before, so the C:N ratio must be affected even if it is not significantly.

L222: I suggest to rephrase: ….did not correlate with any other variable. Because if you state compost has an effect on X and Y you already interpret, which belongs to the discussion part.

L223: Rephrase to avoid the aforementioned interpretation issue: "In the drought treatment aboveground biomass was decreased…. " Check throughout the results chapter, in case I missed a sentence. Also, was this reduction significant?

L225: reduced bulk density

L227: unclear, is this now for the compost x drought plots or all compost plots? And slightly higher means insignificantly? If so add a P<0.05 to be clear on this.

L238: as mentioned before, rephrase to avoid interpretation. In the drought treatment we observed an increase in root tissue density…..

L238-239: Too fuzzy, be more specific. This increase was in all drought treatments i.e. drought and drought x compost?

L244: Sometimes topsoil is 0-5cm, 0-15, 0-30 and here now 5-10cm. To be more coherent, I suggest to stick to the layers, and if the topsoil is meant, always consider the 0-30cm (or however you want to define it, traditionally it would be 0-30cm, some use 0-15, others 0-25). Please check and adjust in the whole manuscript.

L244: correlation in which direction? Positively? Negatively? Please add.

L246: correlation in which direction? Positively? Negatively? Please add.

L249: what is "constant"?

L250 "indicated" Either all in present tense or all in past tense. Avoid to switch, keep it consistent. Also what are "both groups"? For consistency stick with treatments)

L250-252: This is an interpretation and belongs to the discussion part.

L259-260: This, after "suggesting" is an interpretation and needs to move to the discussion.

L270-273: Changes in which direction? Increases, decreases? The curious reader wants to know this. You give this information from L274, so you can cut everything before, as it gives no information.

L294: add a reference for the respiration loss and put into context if this loss is what was expected or higher or lower than one would expect.

L295: recalcitrant type, what does this entail? Is there a clear definition or recalcitrant compared to what? Be more specific. It would be important to also mention that already lots of C is lost when producing the compost to ensure that the reader know this and does not think that compost is a solution to keep **all** C longer in the soil.

L295: effects on what? Be more specific

L298: this part of the sentence could be cut.

L299: cut the word conclusion.

L306: How does it stress this importance? Unclear, as the sentence before is unlcear.

L307-308: In line 292, you already confirmed your 1st hypothesis. Rephrase there.

L308: too fuzzy. Investment of what? Into biomass? Into C incorporation into different parts of the plant? Be more specific.

L307-312: This could be shortened (does not have to be). All of this is well known and thus does not need to be explained so long.

L312-313: What pool? Too fuzzy. The soil C pool and the vegetation C pool. In which direction was it shifted? Be more specific. Also, why not vegetation pool, if you can distinguish more exactly between below and above ground biomass? Thus, the more "general" part of the sentence could be cut.

L314: here you are in present tense. Above in past tense. Keep it consistent.

L316: did others find that as well or are you the first one? A reference would be good.

L326: such as? E.g. soil ad climate could be named.

L332: I read the exact same sentence in the chapter before. Rephrase to keep it interesting. Also here, add what you mean by "investment". What is invested?

L333-336: Rephrase, too complex of a sentence for easy text flow.

L349: Please be precise, what do you mean by "plant biomass"? below? Aboveground? Both together?

L390-393: can be cut. Not based on data from this study and could be move to the introduction. Here, the reader wants to see what your results could mean.

L399: better: "drought conditions"

---

## Author Response (AR1)

*To the editor,*

*We appreciate this opportunity for revising the manuscript. The introduction has been rewritten following the suggestions given by the reviewers, with details outlined below in response to the comments. We believe that the narrative is clearer and the hypothesis better introduced. We have added in the methods and results sections the information that was previously missing and requested by the reviewers. The discussion is backed by more sources. In the document below, we present the response to each instance of the review process. Kind regards on behalf of all listed authors,*

*Daniela Guasconi*

**Reviewer #1**

*Dear anonymous reviewer,*

*We are thankful for the constructive comments to our manuscript. The majority of the feedback received so far concerns wording, paragraph structure and conclusions drawn. If given the opportunity to submit a revised manuscript, we believe that we can appropriately address each of the comments and substantially improve our work. Below we outline the details of how we have proceeded in each instance. Kind regards on behalf of all listed authors,*

*Daniela Guasconi*

**Major comments:**

- The title is misleading. The main topic of the text is about compost and drought effects. The spatial and temporal variability in soil and vegetation is only mentioned a few times and the hypothesis also focus on the compost and drought effects.

*We propose a new title: "Experimental drought and soil amendments affect grassland above- and belowground vegetation but not soil carbon stocks"*

**Abstract**:

- L9: grassland soils have no large C sequestration potential, only a change of management from arable land to grassland has this C sequestration potential. Grassland itself not. Grassland itself is either in a C equilibrium if it is a grassland for long enough or C accrual may happen which may sequester C depending on how this accrual is achieved. Only a global net increase in C is sequestration, otherwise if e.g. compost is used, the C contained in this composed is merely moved from site 1 to site 2 and does not achieve a net uptake of C from the atmosphere into the soil.

*The formulation was changed to "and targeted grassland management has the potential to increase C sequestration potential" (L 8-9)*

- L14: Vegetation C pools are not shown. Biomass is shown.

*We agree with the correction here and in the rest of the text, and propose to change the formulation to "soil C and plant biomass"*

- L14-15: Too fuzzy. You could say that in the compost addition treatment you found higher aboveground biomass. There is a correlation, but you did not prove the causation. Also increased compared to what? The control? Mention this to be clear.

*We propose to change this sentence to "Aboveground biomass and soil C content (% C) increased compared to controls in compost-amended plots" (L 15-16)*

- L16: decreased it only compared to compost treatment. Compared to the control nothing happened.

*We propose to change this sentence to "Drought did not decrease plant biomass compared to control plots"*

- L17-18. Not true based on your data. Soil amendment shifted C allocation upwards as shoot biomass increased, yes. But drought had no effect on belowground biomass. Only the type of roots was affected, not the biomass. Thus, drought had no C shift as a consequence.

*We agree with the correction here and in the rest of the manuscript, and propose to change this formulation to "the soil amendment shifted C allocation to aboveground plant organs."*

- L21: Nowhere, in the complete manuscript do you talk about upscaling. This part needs to be cut

*We cut this part in the abstract.*

**Introduction**:

- L31: "…mitigate soil organic carbon loss…" I would add here that this is the case locally. E.g. mitigate soil organic carbon loss on a specific site. This is relevant, as leakage needs to be taken into account as all amendments are basically biomass from another site, transformed and then brought to the specific site where you want to amend this biomass. You export biomass from site A, reducing C inputs there, and import it to site B, increasing C there. Also, leakage should then be explained to the readers, so they can follow. This matters as this is the major limitation to such

amendments. It needs to be additional biomass globally, not locally, to enable C sequestration in soils by increasing SOC accrual.

*This can be specified here and in the rest of the text, as "mitigate soil organic carbon loss in specific sites" (L 30). In general, we propose to review this argument throughout the text with reference to Moinet et al. (2023) and Don et al (2024). We also clarified the spatial aspect in lines 46-47. When reformulating the hypotheses, we also wish to clarify that one of the goals was to test whether there is a net increase in C which exceeds the mass of C added in the compost (as found in Ryals and Silver, 2013).*

*Moinet, G. Y. K., Hijbeek, R., van Vuuren, D. P., & Giller, K. E. (2023). Carbon for soils, not soils for carbon. Global Change Biology, 29, 2384–2398. https://doi.org/10.1111/gcb.16570*

*Don A, Seidel F, Leifeld J, Kätterer T, Martin M, Pellerin S, Emde D, Seitz D, Chenu C. 2024. Carbon sequestration in soils and climate change mitigation—Definitions and pitfalls. Global Change Biology 30:e16983. doi:10.1111/gcb.16983*

*Ryals, R., & Silver, W. L. (2013). Effects of organic matter amendments on net primary productivity and greenhouse gas emissions in annual grasslands. Ecological applications : a publication of the Ecological Society of America, 23(1), 46–59. https://doi.org/10.1890/12-0620.1*

- L86: The text before does not highlight enough the time scales of SOC formation. Sometimes you refer to land use history which indirectly shows that the effects are decadal up to centennial and more. However, then you conduct three year experiment to search for SOC stock changes which is a very short time for C stabilization experiments. This needs to be mentioned in the discussion the latest. Of course, you find an increase in SOC, as you add C rich material. This is not surprising. But how much of this would be left after 10 years, especially of a one time only amendment which is also not the agricultural reality. This is a drawback of the study that needs to be discussed in the discussion part which it currently is not.

*This aspect was developed further in the first paragraph of the discussion, with reference to Moinet et al (2023) regarding timescales of SOC formation and to studies reporting increased SOC after a single amendment application (Ryals et al. 2013) or after two years (Gravuer et al. 2023). We also consider the comparison between the measured change in total soil C with the remaining compost-derived C in the same soil layer (Fig 1), which is lower. This suggests that the increase in soil C is not only derived from the amendment itself.*

*Ryals, R., & Silver, W. L. (2013). Effects of organic matter amendments on net primary productivity and greenhouse gas emissions in annual grasslands. Ecological applications : a publication of the Ecological Society of America, 23(1), 46–59. https://doi.org/10.1890/12-0620.1*

*Moinet, G. Y. K., Hijbeek, R., van Vuuren, D. P., & Giller, K. E. (2023). Carbon for soils, not soils for carbon. Global Change Biology, 29, 2384–2398. https://doi.org/10.1111/gcb.16570*

*Gravuer K, Gennet S, Throop HL. Organic amendment additions to rangelands: A meta-analysis of multiple ecosystem outcomes. Glob Change Biol. 2019; 25: 1152–1170. https://doi.org/10.1111/gcb.14535*

- L81-89: This belongs partially in the methods and in the results part and not in the introduction.

*The content of this section was moved to the methods.*

- L23-90: The introduction in general, needs to be more focused. You start with general topics, good. Then you need to narrow everything down to the question: Why do we need to study organic amendments more? Show clearly the knowledge gaps and show ways to answer these. These ways will be your hypothesis. This link needs to be clearer (I address this in the specific comments section). In the end, when the readers read your introduction, they need to be able to not only follow your content but also to reach the same conclusion about why does your topic matter and how to fill the knowledge gaps. As of now this is too weak and not focussed enough. When I read your hypothesis, I wonder why you look at organic amendments and not mineral amendments or other management practices. It still seems a little bit "random". With a little bit of work, this can be changed and improve the overall quality of the paper.

*We agree that the hypotheses can be made clearer. Our rationale is that C amendments are being proposed as a generic solution for SOC accrual or even sequestration (Ryals et al. 2015; DeLonge et al. 2013), but the net effects are uncertain and context dependent (Moinet et al. 2023). Compost is even used in some cases as part of C credit trading plans (https://marincarbonproject.org/carbon-farm-plans/). We propose to narrow down the focus and highlight the following contributions of our study: (1) contribute to increase our knowledge of C amendments in diverse land management contexts, in this case old agricultural fields converted to grasslands in Sweden, (2) increase our sampling resolution by sampling at several depths, (3) combine the treatment with drought, to test interactions, (4) test the effect of the treatments on plant biomass both above-and belowground, since the belowground dimension is often ignored. We will also properly introduce the use of compost in relation to previous studies and to our approach of using compost made from a C4 plant that allows us to trace the C isotopes.*

*Moinet, G. Y. K., Hijbeek, R., van Vuuren, D. P., & Giller, K. E. (2023). Carbon for soils, not soils for carbon. Global Change Biology, 29, 2384–2398. https://doi.org/10.1111/gcb.16570*

*Ryals, R., Hartman, M.D., Parton, W.J., DeLonge, M.S. and Silver, W.L. (2015), Long-term climate change mitigation potential with organic matter management on grasslands. Ecological Applications, 25: 531-545. https://doi.org/10.1890/13-2126.1*

*DeLonge, M.S., Ryals, R. & Silver, W.L. A Lifecycle Model to Evaluate Carbon Sequestration Potential and Greenhouse Gas Dynamics of Managed Grasslands. Ecosystems 16, 962–979 (2013). https://doi.org/10.1007/s10021-013-9660-5*

- L91-94: Hypothesis 1: How does compost increase plant growth? Make the mechanism clearer in the introduction (line 58 needs more information on this for the readers to follow). Also, how does compost increase soil C? You did not state that compost is an organic amendment and you did not state that organic amendments are having high C content. Needs to be added in the introduction. L37 states this indirectly which is too fuzzy. Make this connection clearer by explaining e.g. how compost is made and its properties. Then this part of the hypothesis becomes very easy to follow. It remains also too fuzzy how a decrease in bulk density causes a negative effect on C stocks. Everything up to here is known and not suitable for a hypothesis, the real hypothesis follows now "we expect that these mechanisms have counteracting effects on net soil C storage;" This hypothesis is not clear and too fuzzy. Which is the direction of the effect you expect based on the introduction part? Do you expect the positive effect to be stronger or the negative one? This would make for a great hypothesis.

*To introduce our hypothesis on the effect of compost on plant growth we developed the topic in lines 46-51. The rationale behind the use of compost in our experiment was clarified in lines 51-53 and 58-59. We agree with the suggestions by the reviewer to improve both information and logical flow, and how these elements can contribute to draw clearer hypotheses. We clarified how bulk density relates to C stocks in lines 52-55 ("By improving soil structure and reducing compaction, compost additions may also reduce soil bulk density. As SOC stocks are calculated by multiplying C concentration by the bulk density, improved management may also lead to net zero effects on C stock despite increased soil C contents"), and present the hypothesis that a decrease in bulk density will counteract a possible increase in total soil C, resulting in no net increase in soil C storage.*

- L94-95: I recommend to turn this around to be clearer in your communication: Drought will have a weak or non-detectable effect on SOC by decreasing both productivity (organic C input) and respiration (microbial decomposition of SOM). In addition: as for hypothesis 1 it is not entirely clear why you study drought effects. You need to stress this more in the introduction as you address there rather the need for SOC increases to improve soil health and that this could happen via amendments. The climate effect is only mentioned in 1 line or so. Stress this point, as it is highly relevant! Furthermore, be more precise. Will drought have an effect or not? If yes, in which direction do you expect it based on what should be contained in the introduction?

*The second hypotheses will be reformulated as suggested ("Drought will have a weak or non-detectable effect on SOC by decreasing both productivity (organic C input) and respiration (microbial decomposition of SOM)"). The rationale behind studying drought effects was*

*developed in lines 74-80. We propose to strengthen the background on drought with reference on drought effects on grasslands, and why we expect compost amendments to reduce drought effects by enhancing soil water retention (Fischer et al. 2019, Kang et al. 2022).*

*M.W. Kang, M. Yibeltal, Y.H. Kim, S.J. Oh, J.C. Lee, E.E. Kwon, S.S. Lee. Enhancement of soil physical properties and soil water retention with biochar-based soil amendments. Sci. Total Environ., 836 (2022), Article 155746, https://doi.org/10.1016/j.scitotenv.2022.155746*

*Fischer BM, Manzoni S, Morillas L, Garcia M, Johnson MS, Lyon SW. 2019. Improving agricultural water use efficiency with biochar – A synthesis of biochar effects on water storage and fluxes across scales. Science of The Total Environment 657:853–862. doi:10.1016/j.scitotenv.2018.11.312.*

**Methods**:

- L132-135: Too fuzzy. What the auger 15cm long and you took the 1st sample 0-15cm. Then you used the same hole for the next 15cm (15-30) and then the same hole for 30-45cm? Please specify how you did it. Also, in case you used the same hole and the auger was 15cm (and not 45cm) long: Do you not think, that the insertion of the 15cm long auger compacts the soil below 15cm and thus you would get much denser samples for 15-30 and 30-45cm?

*To improve clarity, we propose to modify this section as follows: "Samples for soil bulk density were collected with a large fixed volume root auger with a sharpened cutting edge (8 cm diameter and 15 cm in length; Eijkelkamp, The Netherlands). Three 15 cm segments were collected sequentially using the same hole, reaching a total depth of 45 cm. Upon extraction, the cores were cut into 5 cm segments, and the bulk density was determined after drying the samples at 105 °C".*

*Concerning the issue of compaction, we believe that this was the most appropriate method to use in our experimental setup, since it was not possible to collect samples horizontally by digging 45 cm deep soil pits for each plot. While there is a risk of compaction, this rather occurs in the deeper end of the same core, and not the one below. Because of this, we reasoned that the use of a* short *core or shorter segments (15 cm) would actually minimize compaction compared to a continuous core, and the large volume (250 ml) would minimize errors due to soil loss from the corer during sampling.*

- L137-138: In what kind of increments or is this a bulk sample from 0-100cm? Please add.

*Samples for total C and N and δ13C were collected in 5cm increments. This will be added to the text (L 154).*

- L140: Was this measured pH the actual or the potential pH? Was measured in destilled water or CaCl2? Please add.

*The soil pH was measured in a commercial lab using distilled water (added to the text). Regarding the comment on soil nutrients in the general comments section, it's unclear what kind of information should be added to the methods section.*

- L147: How did you place the auger? On the plants, in between plants? This will be changing the results strongly, so please be more specific on how you did this.

*Because the samples were taken on a grassland with a rather large auger (8 cm in diameter), it was not possible to place it between the plants, only on top. Living aboveground plant biomass and fresh litter however were removed and not included in the soil samples, so it would likely have very little effect on the results. This will be specified in the text. (L 164-167)*

- L148: Why do you sample all plots to 30 cm and then just a subset of 16plots to 45cm? Which were the 16 plots? Which plots and why were they not sampled to 45cm? Please be clearer on this.

*The roots in our study system are very shallow, and we estimated that 95% of the total root biomass in the plots was < 30 cm. The amount of roots in the deep samples (30 – 45 cm) was so low that we considered those samples to be a control for maximum rooting depth. Considering the difficulty of the sampling in the clay-rich soil, especially in the late summer when root samples were collected, we chose one plot per treatment and location for deep root sampling. This can be specified in the text.*

- L157: How did you calculate the stocks? Explain and state which factors you included. Did you consider the packing density? For this, see publications of Poeplau et al, which you already cited for the d13C calculations. He explains how to calculate stock correctly.

*The C stocks were calculated as C content x bulk density x soil volume as was moved to the methods (L 183-191). Therefore, we did account for soil bulk density in the stock calculation, at both the start and end of the experiment, but we did not account for any potential compaction of the soil due to our treatments. Our replicated sampling approach is thus well designed to measure changes in soil properties of specific depth layers. However, we recognize that for analyses of total soil C stock change over time (i.e. a resampling approach comparing change from 2019-2022) one could instead calculate C stock change using an equivalent soil mass procedure (Wendt and Hauser, 2013). We added a figure for this in the supplements.*

*Wendt, J.W. and Hauser, S. (2013), An equivalent soil mass procedure for monitoring soil organic carbon in multiple soil layers. European Journal of Soil Science, 64: 58-65. https://doi.org/10.1111/ejss.12002*

- L168-170: What do you mean by "This is because this soil depth contains the majority of […] the microbiological activity […]" What is a majority of activity? How did you measure this? You have not explained anything about microbiological measurements so I wonder how you could back you sentence up by data. Please adjust.

*While we do have data about total microbial DNA sequences (collected for another study), this sentence as formulated at the moment is misleading, and we propose to modifying by keeping only mention of the root biomass.*

- L174: what was you alpha? 0.05, 0.01? Please add.

*The alpha was 0.05, it will be added. (L 201)*

**Results**:

- L211: Figure 1 does not include the effect of the drought treatments. It would be nice to see them too in a similar figure, especially, as the text above L205-209 does not read well and takes some time to see in which treatment the C derived from compost is higher.

*The figure was modified to present both compost and drought treatments, and their interaction.*

- L232: Figure 2: Why is the 0-5cm box always yellow? Not explained. If there is no reason, I'd suggest to remove this background colour. Problematic is also that there is a dot in the box-plots for the mean value but the points outside the box-plots are outliers. If possible, adjust this to two different symbols.

*The color was meant to differentiate between the size of the first sample (5 cm interval) and of the other three (15 cm intervals). This was now specified in the caption, and the mean value dots changed into a different shape in this and in the other plots.*

- The caption needs to be adjusted: there are no bars, these are called box plots or Box-Whisker-Plots. Please adjust this. Also explain in the caption what C is. Figures must be self-explanatory without the main text. So, the reader needs to learn again that C = carbon

*The caption was modified as suggested in this and in the other figures.*

- L263: Figure 3 similar to above. The yellow background, the C=carbon, the inside and outside dots. In addition, the sampling depths need to be named in the caption. What does "low" and "high" in the sites name mean? Too fuzzy, I assume it is the

upslope and downslope area? In the text you usually call higher position in the catena (or similar), therefore, keep it consistent and call it here the same.

*The low and high refers to low and high catenary positions. The captions were changed, and we will make sure to keep consistency with the terminology in the text.*

- L280: Figure 4: comments as above. In addition: Be more specific as you state (excluding treatment plots). Better write Values of X and Y …..in 2022 of all control plots. Also give us an n= ? How many control plots are meant? This should be a standalone figure. This n in addition, should be also added for all figures above. This helps the reader to understand the values better as it matters if those values are derived from 3 plots or from 100.

*The caption can be modified as suggested with added information.*

- L185 Tabe1: orange and red are hard to distinguish. Also, people with a red-green weakness could not see any difference in colour. Depth with a capital D, Factor with F to keep the table headings consistent. Differences: Add the p value for that. This needs to be a stand alone table. Where do I see the sites names? In the figure before the sites had names, here not. Keep it consistent. Why are the horizontal lines around "year" thicker than all others? If I read catenary position as factor, by just looking at this table and nothing else, I do not know which positions you had, same for all other factors.  Name these.

*We propose to remove the table from the text and place it in the Supplementary material (with the suggested changes). There is no p-value as the effect size is only a measure of the magnitude of difference, not an estimate of probability.*

**Discussion**:

The whole discussion needs to be checked if all statements about the results mentioned here, are actually found in the results. Comments on this will follow here.

- L297: The effect of compost after the few applications is indeed expected to be hardly detectable.  Here you could add results from long-term experiments showing after what time a significant change is to be expected.

*Appropriate references can be mentioned here and in the introduction where they can help support our motivations for the design of the experiment and hypotheses, as suggested in the previous comments. There are several studies that report increased SOC after < 4 years (see Gravuer et al. 2019, Liu et al. 2016), but also long-term studies like Ryals et al. (2015), or mechanisms in Wang et al. (2022)*

*Gravuer K, Gennet S, Throop HL. Organic amendment additions to rangelands: A meta-analysis of multiple ecosystem outcomes. Glob Change Biol. 2019; 25: 1152–1170. https://doi.org/10.1111/gcb.14535*

*Liu, S., Zhang, Y., Zong, Y., Hu, Z., Wu, S., Zhou, J., Jin, Y. and Zou, J. (2016), Response of soil carbon dioxide fluxes, soil organic carbon and microbial biomass carbon to biochar amendment: a meta-analysis. GCB Bioenergy, 8: 392-406. https://doi.org/10.1111/gcbb.12265*

*Ryals, R., Hartman, M.D., Parton, W.J., DeLonge, M.S. and Silver, W.L. (2015), Long-term climate change mitigation potential with organic matter management on grasslands. Ecological Applications, 25: 531-545. https://doi.org/10.1890/13-2126.1*

*D. Wang, J.Y. Lin, J.M. Sayre, R. Schmidt, S.J. Fonte, J.L.M. Rodrigues, K.M. Scow. Compost amendment maintains soil structure and carbon storage by increasing available carbon and microbial biomass in agricultural soil – A six-year field study. Geoderma, 427 (2022), Article 116117*

- L301-302: This conclusion is not entirely correct. It may be part of the reason, surely the climate of the growing season differed as well to have an effect. For the suspected increase in N (which you did not measure) I suggest to add a reference showing this. In these lines be a bit more careful, as compost is one of multiple reasons why the biomass is be increased as you cannot prove that it was the compost treatment alone.

*Total soil N was measured (line 156) and reported (line 249). We propose to reformulate the sentence as "the significant increase in aboveground biomass three years after the compost application could partly be explained by the persistence of favorable plant growing conditions, such as increased N in the soil. This mechanism was invoked by Oladeji et al (2020), and may interact with precipitation-related interannual variability in plant growth (Sala et al. 2012)"*

*Sala Osvaldo E., Gherardi Laureano A., Reichmann Lara, Jobbágy Esteban and Peters Debra 2012 Legacies of precipitation fluctuations on primary production: theory and data synthesisPhil. Trans. R. Soc. B367: 3135–3144. http://doi.org/10.1098/rstb.2011.0347*

*Oladeji, O., Tian, G., Lindo, P., Kumar, K., Cox, A., Hundal, L., Zhang, H. and Podczerwinski, E. (2020), Nitrogen release and plant available nitrogen of composted and un-composted biosolids. Water Environ Res, 92: 631-640. https://doi.org/10.1002/wer.1260*

- L302-305. Cut into two sentences to improve understanding. Also, what do you mean that compost application "was manifested as extended growing season"? I cannot follow the interpretations here.

*This section was changed to (L332-336): "Our results suggest that compost treatments might benefit the ecosystem C balance indirectly through increased biomass production, such as in this case, or by extending the growing season, such as in Fenster et al. (2023). These interactions between land management, vegetation growth and plant-derived C inputs also underline the importance of including vegetation dynamics when assessing the effectiveness of C management".*

- L321: This is no C sequestration. This is C accrual. C sequestration in soils is always a global net increase. Compost is biomass exported from another area of land and imported on this plot. That entails a C loss in site 1 and an increase in site 2. Thus, just a shift of global stocks, not an increase. This is important as currently the discussion about the possibility of agriculture to offset GHG emissions is heated and lots of impossible hopes are there. Therefore, I strongly suggest in order to keep the communication clear, do not call compost amendments a C sequestration measure or similar. This is merely a management option to shift C from one site to another to improve soil health. C sequestration however is about offsetting climate change effects. In addition, as you state that there are no increases in soil C, even the term C accrual does not fit well either. Thus, better use potential C accrual.

*We agree on the corrections related to terminology, here and in the rest of the manuscript. We propose to change this sentence into: "the potential C accrual brought by the increased plant productivity…". In general, we propose to review the manuscript after the terminology suggested by Don et al. (2024). We do wish to note however, that if net soil C stocks increase by more than the C mass added via compost (as described by eg. Ryals et al 2013), that does constitute C sequestration.*

*Don A, Seidel F, Leifeld J, Kätterer T, Martin M, Pellerin S, Emde D, Seitz D, Chenu C. 2024. Carbon sequestration in soils and climate change mitigation—Definitions and pitfalls. Global Change Biology 30:e16983. doi:10.1111/gcb.16983*

- L325: Also here, be careful with the term C sequestration. Here, I suggest to use …seeking to increase C accrual (or C stocks)….

*The sentence can be modified as per suggestions*

- L327: How can the Tovetorp grassland drive the increase in aboveground plant biomass? Unclear and fuzzy. What is a grass rich plot? Unclear. How can you make a statement about effects of plant community composition when you never measured it? The fact that C in the 0-5 cm plot was higher in that site could also have an effect, as you basically want to state more Soil C = more biomass. This whole conclusion here is not convincing yet and needs to be revised and sharpened. What is it that you want the reader to take as a take home message from this chapter? I could not tell yet. The last sentence here could for example be a statement/conclusion in 1 sentence about the compost effect on soil C (stocks?) and plant growth based on what you said before in

this chapter. This (as of now lacking) key message would also allow the reader to better follow then the drought effects chapter.

*We agree that the current formulation is unclear. Plant community composition within the same plots was measured and reported in Roth (2023), including the relative abundance (in biomass) of grasses and forbs. Our rationale is that the dominant plant functional group might be important in determining the effects of the amendments (i.e. grasses might be benefited more than forbs after compost addition). Similarly, we hypothesize that the higher C in that site could be a function of the plant community composition (discussed in Roth, 2023). This was also found in Ryals et al. (2016).*

*The paragraph was re-elaborated and we hope that the message is now delivered in a more clear manner.*

*We modified the section as following: "For instance, the effect of compost applications on plant biomass can also be a function of plant community composition, as compost addition may favor grasses over forbs (Ryals et al. 2016). In our experiment, aboveground biomass increase was highest at the site with the greatest abundance of grasses (table S4), suggesting that grassland management schemes should consider interactions with plant community composition." (L358-362) and added a table describing plant community composition in the supplements.*

*Ryals R, Eviner VT, Stein C, Suding KN, Silver WL. 2016. Grassland compost amendments increase plant production without changing plant communities. Ecosphere 7:e01270. doi:10.1002/ecs2.1270*

- L333-336: This statement in these lines is correct. Now take this into account when discussing my comment above for L301-302.

*Noted, this was mentioned now also in lines 331-332*

- L338: Does your weather data availability allow to also look at the distribution of rainfall along the seasons? As you mention it in L333-334. That would be helpful.

*Yes, we do possess daily and monthly precipitation data for the study years. We suggest to add a table with monthly precipitation data 2018-2022 to the Supplements.*

- L333-348: This paragraph could move to materials and methods as the site description. Here, you do not discuss your drought treatment effect and thus this section does not really fit in here.

*We propose to move the information on precipitation for the study years to the methods, and refer to it in the discussion when discussing legacy effects of precipitation.*

- L347-348: This is purely hypothetical and you have no reference nor data on this. I suggest to add references with a statement about how much this could in similar situations affect the data you have. If I only read the text as is, I could think that the result of the experiment is pure coincidence and that we could learn nothing from this data (which is not true!). Thus, put it into context and add a number about how much e.g the drought effects could be weakened.

*We propose to discuss this hypothesis in relation to relevant literature, for example Sala et al. (2012) or Reichmann and Sala (2014)*

*Sala Osvaldo E., Gherardi Laureano A., Reichmann Lara, Jobbágy Esteban and Peters Debra 2012 Legacies of precipitation fluctuations on primary production: theory and data synthesisPhil. Trans. R. Soc. B367: 3135–3144. http://doi.org/10.1098/rstb.2011.0347*

*Reichmann, L.G. and Sala, O.E. (2014), Differential sensitivities of grassland structural components to changes in precipitation mediate productivity response in a desert ecosystem. Funct Ecol, 28: 1292-1298. https://doi.org/10.1111/1365-2435.12265*

- L349-351: How so? I cannot follow how plant community fits in here and how this is linked to your results, as you did not measure plant community structure

*We propose to remove this speculation as less relevant for our specific setup, and mention only interannual variation and potential methodological issues of the experimental treatments.*

- L351-353: Where are those results?

*The sentence in its current state is misleading, we propose to mention only the part relevant to our study, i.e. that it was not possible to link plant functional group and root biomass.*

- L362-372: And how is this linked to your study and results? Be precise, do not imply it, state it and name the link. Also, this whole paragraph could be shortened and condensed.

*We see value in mentioning the limitations of the experimental setup, especially seen that we expected a reduction in plant biomass under drought. However, we agree that the paragraph could be shortened and condensed. We propose to mention the main limitation of the standardized partial rainout shelter design (i.e. the precipitation that reaches the ground might be enough to not cross a threshold of water availability stress for the plants), and the main ways experimental drought in the field differs from natural droughts.*

- L373-379: How does this link to your study and why is it relevant? The second half from L379 on is relevant, this part could be cut.

*We see a value to the reader that mechanisms driving C stocks changes identified in other studies are exemplified. We would like to retain this text, although it was streamlined.*

- L379-388: You mentioned already before, that your experimental drought may not be intense enough, and you focus again on microbial activity. This whole chapter reads not like you explaining what you found/measured/observed but rather like: We did not find anything and here are the "excuses". This chapter needs rephrasing. Focus less (but mention!) what may have gone wrong (e.g. drought intensity) and then talk about what you found and what your data allows for interpretation. In addition, C storage in soils is a process that takes longer time scales than the time interval you measured. E.g. if you change your agricultural management (i.e. introducing cover crops to arable land) the C stocks may increase (through increased biomass and thus C input). This increase is not measureable after 3 or 4 years and not with so few samples as you have. E.g. for soil C monitorings timesteps of 10 years and thousands of samples are needed to identify reliably if C stocks actually did increase. This is a longterm process and thus I am not surprised that you did not find many differences. Apart from adding C via manure, which of course immediately increases the C stocks.

*Although we see value in mentioning the role of microbial activity, we agree with the suggestion of reformulating the chapter in a more focused way. As mentioned in a comment above, we also propose to cite studies reporting increased SOC after < 4 years (see Gravuer et al. 2019, Liu et al. 2016), but also long-term studies like Ryals et al. (2015), or mechanisms in Wang et al. (2022). The paragraph has been partly restructured.*

*Gravuer K, Gennet S, Throop HL. Organic amendment additions to rangelands: A meta-analysis of multiple ecosystem outcomes. Glob Change Biol. 2019; 25: 1152–1170. https://doi.org/10.1111/gcb.14535*

*Ryals, R., Hartman, M.D., Parton, W.J., DeLonge, M.S. and Silver, W.L. (2015), Long-term climate change mitigation potential with organic matter management on grasslands. Ecological Applications, 25: 531-545. https://doi.org/10.1890/13-2126.1*

*D. Wang, J.Y. Lin, J.M. Sayre, R. Schmidt, S.J. Fonte, J.L.M. Rodrigues, K.M. Scow. Compost amendment maintains soil structure and carbon storage by increasing available carbon and microbial biomass in agricultural soil – A six-year field study. Geoderma, 427 (2022), Article 116117*

- L396: What do you mean "slightly"? Significantly? If not, then better state "tended to reduce"

*We will change to "tended to reduce"*

- L397: Figure 4 is not showing this. There you see the treatments excluded. In Figure 2 you see the treatments. There is no difference in R:S ratio. This needs to be corrected and changed.

*The figure number is an oversight, and will be changed to figure 2. The formulation is also misleading, and we propose to change it to "there is a tendency for higher root:shoot ratio in the plots with combined compost and drought treatment (Fig. 2)".*

- L397-403: Here, we would need references, if any of this will remain after the reanalysis of the comment of line 397.

*The section on root:shoot ratio will be revisited and appropriate references will be added if needed, for example Eziz et al. 2017, Guswa (2010) (roots and drought).*

*Eziz, A., Yan, Z., Tian, D., Han, W., Tang, Z., Fang, J., 2017. Drought effect on plant biomass allocation: a meta-analysis. Ecol. Evol. 7 (24), 11002–11010.*
*https://doi.org/10.1002/ece3.3630.*

*Guswa, A.J., 2010. Effect of plant uptake strategy on the water− optimal root depth: effect of plant uptake strategy on root. Water Resour. Res. 46 (9)*
*https://doi.org/10.1029/2010WR009122*

- L406: vegetation C content was one of the goals? Then why was there no data on the C content of the plants? Cut this part or rephrase.

*This part will be reformulated as "biomass"*

- L 407-408 No it did not. The changes in roots are insignificant and there is not even a pattern, for shoots you are right.

*We meant to say that shoots were affected but roots were not. We propose to reformulate to "The compost treatment led to an increase in biomass in shoots but not in roots".*

- L408-409: No, it did not. Fig 2 shows that shoot biomass in the compost treatment was different from all other treatments.

*We propose to reformulate this as: "Drought did not decrease plant biomass, but led to shifts in root traits"*

- L410: Where do you see these changes in C allocation? Not in Figure 2.

*We propose to reformulate this as: "Compost amendment and drought had distinct effects on plant shoot and root growth"*

- L411: How do they improve our understanding? Tell us.

*We propose to complete this formulation with: "by illustrating the different components of plants and soil properties affected by the management"*

- L412: How can your findings contribute to improve modelling? Unclear and not discussed in the discussion part at all. Thus, this cannot be concluded.

*This sentence will be removed.*

- L414: Why do you leave out the part about the effect of precipitation which you mentioned in the discussion?

*This is a good suggestion, it will be added. (L 439)*

- L415-416: Yes of course. But this has been known before and is nothing new found by your study. This sentence could have been stated like this even without your study and thus does show the value of your study.

*This sentence will be removed.*

**Specific comments:**

L23-24: add some sources e.g. 4permille initiative, EU green deal, Farm to fork. Anything to show this increased interest

*The 4 per mille initiative is mentioned in line 33, but we can add appropriate references here as well.*

L27: "Sometimes": This makes me wonder, what it is called other times. Either cut the sometimes or give 1 or 2 more other names

*We propose to change the formulation to "generally" and add an appropriate reference, e.g. Paul et al. (2023)*

*Paul C, Bartkowski B, Dönmez C, Don A, Mayer S, Steffens M, Weigl S, Wiesmeier M, Wolf A, Helming K. 2023. Carbon farming: Are soil carbon certificates a suitable tool for climate change mitigation? Journal of Environmental Management 330:117142. doi:https://doi.org/10.1016/j.jenvman.2022.117142*

L33-36: This is basically, what C sequestration in soils means. I would move this up and use the term C sequestration in soils thereafter

*We can modify the sentence as suggested*

L 34-43: You are right with what you say and here we need some literature references to back the mentioned topics up.

*We propose to add appropriate references, e.g.:*

*Paul C, Bartkowski B, Dönmez C, Don A, Mayer S, Steffens M, Weigl S, Wiesmeier M, Wolf A, Helming K. 2023. Carbon farming: Are soil carbon certificates a suitable tool for climate change mitigation? Journal of Environmental Management 330:117142. doi:https://doi.org/10.1016/j.jenvman.2022.117142*

*Don A, Seidel F, Leifeld J, Kätterer T, Martin M, Pellerin S, Emde D, Seitz D, Chenu C. 2024. Carbon sequestration in soils and climate change mitigation—Definitions and pitfalls. Global Change Biology 30:e16983. doi:10.1111/gcb.16983*

*Almaraz M, Simmonds M, Boudinot FG, Di Vittorio AV, Bingham N, Khalsa SDS, Ostoja S, Scow K, Jones A, Holzer I, Manaigo E, Geoghegan E, Goertzen H, Silver WL. 2023. Soil carbon sequestration in global working lands as a gateway for negative emission technologies. Global Change Biology 29:5988–5998. doi:10.1111/gcb.16884*

L37: "directly increases the standing stock of SOC" instead of "standing" I suggest to use "local" or "SOC stock of the site"

*We propose to modify the sentence as suggested*

L42: Wording. The term "Soil C sequestration" is currently under debate as it suggests soil carbon is sequestered when in fact atmospheric carbon is being sequestered in soils. Thus, I suggest to write "potential C sequestration in soils"

*We propose to modify the sentence as suggested*

L43: ….contributions to soil C stocks.

*Noted*

L43-44: You could also mention that roots are 2/3 more recalcitrant to decomposition and thus of a different quality that above ground biomass and thus better suited when trying to sequester C in soil

*Agreed, we propose to add this with reference to:*

*Rasse DP, Rumpel C, Dignac MF. 2005. Is soil carbon mostly root carbon? Mechanisms for a specific stabilisation. Plant and Soil 269:341–356. doi:10.1007/s11104-004-0907-y*

*Gaudinski JB, Trumbore SE, Davidson EA, Zheng S. 2000. Soil carbon cycling in a temperate forest: radiocarbon-based estimates of residence times, sequestration rates and partitioning of fluxes. Biogeochemistry 51:33–69. doi:10.1023/A:1006301010014*

L44-47: Please add a reference or two

*We propose to add reference to:*

*Hayes MA, Jesse A, Tabet B, Reef R, Keuskamp JA, Lovelock CE. 2017. The contrasting effects of nutrient enrichment on growth, biomass allocation and decomposition of plant tissue in coastal wetlands. Plant and Soil 416:193–204. doi:10.1007/s11104-017-3206-0.*

L52: "whether through conventional or regenerative methods" I would cut this part. It does not add any information and if you mention it you should clarify what "regenerative" methods are and what not. To avoid getting side tracked, I suggest to cut this.

*We propose to cut the part, as suggested*

L52-54: Already mentioned e.g. in L25-26. Redundant. Rather make a bridge from the last paragraph over to amendments.

*The repetition of the concept is an oversight, we propose to change the formulation*

L63: see my specific comment on line 42

*Noted*

L62-68: but this is also true for cropland. Why only consider grassland here when you talked about all agricultural land before?

*Our intention was to narrow the focus on grasslands since it is where our study has been carried out. However, we propose to change the wording to improve consistency*

L70: here you go back to crop yields e.g. cropland. Looking at the comment above, you switch randomly between all agricultural land, then specifically crop or grassland and back to all agricultural land. I suggest to harmonize it and talk about all agricultural land as long as possible until you get really into the details of grasslands and the topic you want to focus on.

*As per comment above, we agree with the suggestion and will reformulate the reference to agricultural land vs grassland in the introduction*

L80: Agreed. But why on grassland? Stress the lack of data there more.

*This can be added to the introduction, in accordance with the comments above*

L86: Their? What does this refer to? The yearly droughts or the yearly droughts and the compost application? Fuzzy, please specify

*This was a typo, the sentence should be read as "the effects"*

L87: soil organic C measured and soil C stocks. Please explain to the reader how SOC and C differ from one another and how these are linked. If I know little about soil C I would wonder how you measure SOC and then derive C stocks and not SOC stocks. Too fuzzy and needs to be cleared up to ensure that everyone understands what you did.

*We propose to reformulate the section and explain the difference between C and SOC.*

L90: Why did you hypothesize this? This needs to be better linked to the introduction text.

*The introduction will be revised as per the suggestions made in the "major comments" section.*

L99-102: Move to M&M

*This part can be moved as suggested*

L108: wording…I suggest: Today, the land management consists of grazing and hay production.

*We agree with the suggestion*

L112: It would be great to know the soil type for the 4 sites, e.g. are we talking about Cambisol and a more exact texture would be great, if available.

*We can add more details about the texture*

L115: Mention the value so we can see the difference.

*The value can be added here*

L157-159: Consider to split this sentence into two. I needed to read it 3 times to follow along.

*We propose to change the sentence to: "Total C content was calculated for all 48 plots as % of total mass, and C stocks were calculated using total C content, bulk density and SOM data. These were then normalized by soil sample thickness (kg/m3)"*

L167: "compost or drought-no compost)" add here a "respectively" after between "compost" and ")".

*Noted*

L184: "the C isotopic ratio" better delta 13C ratio. Be consistent.

*Noted*

L185ff: What is this "model"?

*We refer to the mixed linear model, this will be specified*

L187: "Landscape variability" Do you mean the position in the slope? Please explain.

*Yes, this will be changed to grassland and elevation (making sure the terms are using consistently in the text)*

L198: I suggest "The drought treatment" to be 100% clear, that this is a treatment and not a "normal" drought. In case the reader skips right to this chapter.

*Agreed*

L200: You mention spring and summer (=seasons) then you mention also growing season. How is this defined, what time is the growing season? Please add.

*This information will be added to the text*

L205-209: It would be informative to see the standard deviation of these numbers as well.

*The standard deviation can be added in the text*

L214: This sentence is already an interpretation which belongs to the discussion part. I suggest to rephrase e.g. In the compost treatment total soil C content and aboveground biomass was increased.

*We propose to change the formulation as suggested*

L218: Was N significantly increased or not? Please add a p value

*We will add this information in the text*

L220: "the C:N ratio" add: significantly. Because N content increases as you stated in the sentence before, so the C:N ratio must be affected even if it is not significantly.

*We will add this information in the text*

L222: I suggest to rephrase: ….did not correlate with any other variable. Because if you state compost has an effect on X and Y you already interpret, which belongs to the discussion part.

*Noted*

L223: Rephrase to avoid the aforementioned interpretation issue: "In the drought treatment aboveground biomass was decreased…. " Check throughout the results chapter, in case I missed a sentence. Also, was this reduction significant?

*We propose to change the formulation as suggested*

L225: reduced bulk density

*Noted*

L227: unclear, is this now for the compost x drought plots or all compost plots? And slightly higher means insignificantly? If so add a P<0.05 to be clear on this.

*We will add this information in the text*

L238: as mentioned before, rephrase to avoid interpretation. In the drought treatment we observed an increase in root tissue density…..

*Noted*

L238-239: Too fuzzy, be more specific. This increase was in all drought treatments i.e. drought and drought x compost?

*We will add this information in the text*

L244: Sometimes topsoil is 0-5cm, 0-15, 0-30 and here now 5-10cm. To be more coherent, I suggest to stick to the layers, and if the topsoil is meant, always consider the 0-30cm (or however you want to define it, traditionally it would be 0-30cm, some use 0-15, others 0-25). Please check and adjust in the whole manuscript.

*We agree and will adjust this in the text*

L244: correlation in which direction? Positively? Negatively? Please add.

*We will add this information in the text*

L246: correlation in which direction? Positively? Negatively? Please add.

*We will add this information in the text*

L249: what is "constant"?

*We meant that the r value did not change. We propose to reformulate as "The strength of the correlation did not differ between control and compost-treated plots"*

L250 "indicated" Either all in present tense or all in past tense. Avoid to switch, keep it consistent. Also what are "both groups"? For consistency stick with treatments)

*Noted, will be changed*

L250-252: This is an interpretation and belongs to the discussion part.

*We can move this part as suggested*

L259-260: This, after "suggesting" is an interpretation and needs to move to the discussion.

*We can move this part as suggested*

L270-273: Changes in which direction? Increases, decreases? The curious reader wants to know this. You give this information from L274, so you can cut everything before, as it gives no information.

*Noted*

L294: add a reference for the respiration loss and put into context if this loss is what was expected or higher or lower than one would expect.

*The sentence was reformulated as "This difference was lower than the estimated C addition (~0.54 kg C m-2) and thus lower than expected, but is, likely due to respiration loss.", and more references are provided when the issue is described more in detail later in the paragraph (L 350-354)*

L295: recalcitrant type, what does this entail? Is there a clear definition or recalcitrant compared to what? Be more specific. It would be important to also mention that already lots of C is lost when producing the compost to ensure that the reader know this and does not think that compost is a solution to keep all C longer in the soil.

*The sentence was reformulated as: "Compost is partly decomposed organic matter, and thus more chemically recalcitrant than fresh grass residues. As a result, its effects on SOC accrual can be persistent over several years (Sarker et al. 2022) and after a single application (Ryals et al. 2013)."*

L295: effects on what? Be more specific

*The specification can be added*

L298: this part of the sentence could be cut.

*Noted*

L299: cut the word conclusion.

*Noted*

L306: How does it stress this importance? Unclear, as the sentence before is unclear.

*This reasoning can be developed in the paragraph*

L307-308: In line 292, you already confirmed your 1st hypothesis. Rephrase there.

*This will be adapted after revising the formulation of the hypotheses in the introduction, as suggested above*

L308: too fuzzy. Investment of what? Into biomass? Into C incorporation into different parts of the plant? Be more specific.

*We propose to change the term with "growth"*

L307-312: This could be shortened (does not have to be). All of this is well known and thus does not need to be explained so long.

*This section can be shortened*

L312-313: What pool? Too fuzzy. The soil C pool and the vegetation C pool. In which direction was it shifted? Be more specific. Also, why not vegetation pool, if you can distinguish more exactly between below and above ground biomass? Thus, the more "general" part of the sentence could be cut.

*We propose to rephrase the sentence as: "this suggests that the compost treatment led plants to preferentially allocate growth to aboveground organs"*

L314: here you are in present tense. Above in past tense. Keep it consistent.

*We will revise the verbal tense throughout the text*

L316: did others find that as well or are you the first one? A reference would be good.

*We are not aware of studies testing root trait response to organic amendments*

L326: such as? E.g. soil ad climate could be named.

*Noted*

L332: I read the exact same sentence in the chapter before. Rephrase to keep it interesting. Also here, add what you mean by "investment". What is invested?

*This was an oversight, the sentence can be rephrased*

L333-336: Rephrase, too complex of a sentence for easy text flow.

*We propose to rephrase the sentence as: "Plant growth is very sensitive to yearly fluctuations and even intra-annual distribution of precipitation (Knapp and Smith 2001, Porporato et al. 2006). Because our analyses are based on only two temporal datapoints (2019 and 2022), it is difficult to assess whether drought reduced plant turnover, defined as the ratio of standing biomass to net primary productivity (NPP)."*

L349: Please be precise, what do you mean by "plant biomass"? below? Aboveground? Both together?

*We propose to add "both above- and belowground"*

L390-393: can be cut. Not based on data from this study and could be move to the introduction. Here, the reader wants to see what your results could mean.

*Noted*

L399: better: "drought conditions"

*Noted*

**Reviewer #2**

Here, results from a three-year field experiment are reported. The study elucidated the interplay between vegetation, soil carbon, drought, and management (i.e. organic amendments) in experimental grasslands in Northern Europe. The study addresses a critical, yet way too often overlooked aspect of soil carbon dynamics in response to disturbances: The effect of the disturbances on plants themselves. Therefore, the study definitely bears merit and reports findings that are important not only to the soil science community but also for ecologists, agronomists, etc. With that being said, I also think that the manuscript in its current form has several major shortcomings that need to be addressed so that the reader clearly understands the "why's" and "how's" of the study.

*Dear anonymous reviewer,*

*We greatly appreciate the constructive feedback provided on our manuscript. Should we have the opportunity to submit a revised version, we are confident that we can address all concerns raised and significantly enhance the quality of our manuscript. Attached below is a detailed plan outlining how we propose to address each comment. Kind regards on behalf of all listed authors,*

*Daniela Guasconi*

General comments

1) The use of "soil health" does not really make sense to me for this manuscript, which entirely focuses on soil carbon. Don't get me wrong, this focus is completely fine (!!!!) but soil health is a much more holistic concept that extends far beyond soil carbon. Hence, using the term "soil health" appears to me to be a hunt for trendy buzz words. I strongly suggest to delete the term or make very clear the soil carbon is a feature of the much more broad soil health concept.

*We propose to review the use of the term "soil health" in the two instances where it appears, deleting "can improve soil health," in L10 and changing in L23 "Management of soil health and soil carbon (C) stocks has been receiving…" into "Soil management has been receiving…"*

2) The title should be adapted to better reflect the story presented in the paper.

*We suggest to change the title from "Spatial and temporal variability in soil and vegetation carbon dynamics under experimental drought and soil amendments" to the more focused "Experimental drought and soil amendments affect grassland above- and belowground vegetation but not soil carbon stocks"*

3) Throughout the text, the authors make rather fuzzy connections between land management, climate change and the mitigation/adaptation of climate change, plant growth, soil health etc. One prime example for this is third paragraph of the Introduction (L52-68) or Fig 2-4 that present the same properties comparing treatments (2), locations (3), and years (4). While all these connections are true and absolutely relevant (not just for the presented study but far beyond), they could (and actually should in my opinion) be presented in a much more concise and clear way. As it is now, it remains kind of unclear how the study is connected to global issues and which exact knowledge gaps the study aims to address. I think a more concise and clear line of argumentation is needed so that the study gets the attention it definitely deserves. I.e. "what is(are) the problem(s), how can they potentially be addressed, what are the knowledge gaps, and how does the presented work close these gaps".

*The main aim of the paragraph in question was to introduce the experimental design, which we wish to retain, as well as the figures. We agree however, that the text can be reformulated as to be clearer and more concise. The hypotheses will further be reformulated as per suggestions from reviewer #1, which should make the whole introduction more consistent with our results and discussion.*

*We will present how C amendments, in addition to their benefits for eg. crop production, are being proposed as a generic solution for SOC accrual or even sequestration (Ryals et al. 2015; DeLonge et al. 2013), but the net effects are uncertain and context dependent (Moinet et al. 2023). To clarify how compost amendment affects SOC at different depths as well as both above- and below-ground vegetation, in this contribution we aim to: (1) increase our knowledge of C amendments in diverse land management contexts, in this case old agricultural fields converted to grasslands in Sweden, (2) increase sampling resolution compared to previous studies by sampling at several depths, (3) combine the compost amendment treatment with a drought treatment, to test interactions, (4) test the effect of the treatments on plant biomass both above-and belowground, since the belowground dimension is often ignored.*

*Moinet, G. Y. K., Hijbeek, R., van Vuuren, D. P., & Giller, K. E. (2023). Carbon for soils, not soils for carbon. Global Change Biology, 29, 2384–2398. https://doi.org/10.1111/gcb.16570*

*Ryals, R., Hartman, M.D., Parton, W.J., DeLonge, M.S. and Silver, W.L. (2015), Long-term climate change mitigation potential with organic matter management on grasslands. Ecological Applications, 25: 531-545. https://doi.org/10.1890/13-2126.1*

*DeLonge, M.S., Ryals, R. & Silver, W.L. A Lifecycle Model to Evaluate Carbon Sequestration Potential and Greenhouse Gas Dynamics of Managed Grasslands. Ecosystems 16, 962–979 (2013). https://doi.org/10.1007/s10021-013-9660-5"*

4) A general question I have is whether the relatively short time period covered in the presented study (3 years) is enough to make any robust conclusions about trends in SOC content and/or stocks? I thought that at least 10 years of continuous treatments are needed to make any sort of conclusions. Hence, I think that this needs to be discussed and conclusions on C sequestration as such (not the potential to foster C seq.!) need to be done with appropriate caution.

*We agree with this point, and, as suggested by another reviewer, we propose to add relevant references to discuss the time frame of C sequestration and of trends in SOC stocks. There are several studies that report increased SOC after < 4 years (see Gravuer et al. 2019, Liu et al. 2016), or six years (Wang et al. 2022).*

*Gravuer K, Gennet S, Throop HL. Organic amendment additions to rangelands: A meta-analysis of multiple ecosystem outcomes. Glob Change Biol. 2019; 25: 1152–1170. https://doi.org/10.1111/gcb.14535*

*Liu, S., Zhang, Y., Zong, Y., Hu, Z., Wu, S., Zhou, J., Jin, Y. and Zou, J. (2016), Response of soil carbon dioxide fluxes, soil organic carbon and microbial biomass carbon to biochar amendment: a meta-analysis. GCB Bioenergy, 8: 392-406. https://doi.org/10.1111/gcbb.12265*

*Ryals, R., Hartman, M.D., Parton, W.J., DeLonge, M.S. and Silver, W.L. (2015), Long-term climate change mitigation potential with organic matter management on grasslands. Ecological Applications, 25: 531-545. https://doi.org/10.1890/13-2126.1*

*D. Wang, J.Y. Lin, J.M. Sayre, R. Schmidt, S.J. Fonte, J.L.M. Rodrigues, K.M. Scow. Compost amendment maintains soil structure and carbon storage by increasing available carbon and microbial biomass in agricultural soil – A six-year field study. Geoderma, 427 (2022), 116117"*

5) If you want to explicitly state hypotheses that were tested (which is a good idea but not needed per se), they need to be rephrased. As they phrased now, they are extremely difficult to grasp and it remains unclear what has been tested with each of the hypotheses. I.e. if you want to included them, they need to be clear and straight to the point (no "we expect that X because of Y if Z is also the case").

*We agree with the suggestion, and the hypotheses will be reformulated. As defined in a comment above: our rationale is that C amendments are being proposed as a generic solution for SOC accrual or even sequestration (Ryals et al. 2015; DeLonge et al. 2013), but the net effects are uncertain and context dependent (Moinet et al. 2023). Compost is even used in some cases as part of C credit trading plans (https://marincarbonproject.org/carbon-farm-plans/). We propose to narrow down the focus and highlight the following contributions of our study: (1) contribute to increase our knowledge of C amendments in diverse land management contexts, in this case old agricultural fields converted to grasslands in Sweden, (2) increase our sampling resolution by sampling at several depths, (3) combine the treatment with drought, to test interactions, (4) test the effect of the treatments on plant biomass both above- and belowground, since the belowground dimension is often ignored. We will also properly introduce the use of compost in relation to previous studies and to our approach of using compost made from a C4 plant that allows us to trace the C isotopes.*

6) It remains unclear how soil samples were taken. How many samples were taken per plot (intact cores for bulk density and loose soil for SOM etc., samples for root biomass)? Hence, it is impossible to know if the values presented are based on robust average values of each plot (which implies that several pseudo-replicates were taken per plot) or not. Similarly, the description for the root phenotyping using WinRhizo is far from complete (e.g. what was the resolution of the scans? was there any filtering used pre and/or post scanning? what kind of scanner was used?). These points are not just details but must be reported accurately and with care.

*To improve clarity, we propose to modify this section as follows (integrating the suggestions from reviewer #1): "Soil and root samples were collected in three replicates from each of the four sites and treatments (one sampling per plot) at the end of the first growing season in 2019 (August - September), and again at the end of the experiment in 2022 (August and October). Samples for soil bulk density were collected with a large fixed volume root auger with a sharpened cutting edge (8 cm diameter and 15 cm in length; Eijkelkamp, The Netherlands). Three 15 cm segments were collected sequentially using the same hole, reaching a total depth of 45 cm. Upon extraction, the cores were cut into 5 cm segments, and the bulk density was determined after drying the samples at 105 °C".*

*Regarding the root image analyses, we propose to integrate the information as follows: "The roots were rinsed with water on a 0.5 mm mesh sieve to remove soil, then placed on a transparent tray, covered with water and scanned with a flatbed scanner at 600 dpi (greyscale), followed by drying at 60 °C for 48 h to obtain the dry weight. The scanned images were analyzed with WinRhizo (Regent Instruments, Québec, CA) to obtain root volume, length and diameter, used to calculate root mass density ($g_{roots}$ $cm^{-3}$ soil), specific root length (cm $g^{-1}$ roots) and root tissue density ($g_{roots}$ $cm^{-3}$ roots)."*

**Minor comments**

L11: I suggest changing to "water retention and infiltration".

*We can modify the sentence as suggested*

L12-13: The part starting with "which" is kind of unclear. Can you split this into two sentences to make it clearer?

*We propose to change the sentence to: "However, soil C dynamics are deeply linked to vegetation response to changes in both management and climate. The responses to environmental change may also be manifested differently in roots and shoots."*

L18-19: This sentence is not well connected to the rest of the abstract. I suggest rephrasing or deleting of the statement.

*We propose to delete the sentence.*

L34-36: To me this is very strange phrasing and not really accurate for your case, i.e. compost addition where the plants are grown somewhere else, and the carbon is then transferred from one place to another where it finally enters the soil. Please rephrase.

*We agree on this point. We suggest to change the sentence to: "Consequently, soil C management via compost amendments aims at transferring plant-derived organic matter to facilitate C accumulation in the soil C pool in specific locations, where it can be retained over long time scales".*

L52-53: What do you mean with "conventional/regenerative" methods? Without any further explanations, these two expressions do not mean anything. In my opinion, this is goes in a similar direction as the use of "soil health", i.e. fancy buzz words are used just for the sake of using them.

*We propose to change the sentence to: "Land management practices – including compost amendments - can significantly impact both above- and belowground plant biomass, which contribute differently to SOC storage."*

L57: The use of the term "element" is misleading as one might think of chemical elements relevant in soil. Please adapt.

*We propose to change this to "...soil microbial communities, plant roots and ecological interactions"*

L58: Similar as above, "nutrients" is not a very accurate term here since microbes and plants use SOM in different ways. Microbes take up the C (and ev. other elements) of the organic matter, while plants may use the nutrients that are mineralised from SOM by microbes. Please be specific/correct and rephrase accordingly.

*We propose to change this to "As microbes decompose this organic matter, they release nutrients in forms that plants can readily absorb (Malik et al. 2013). In turn, the increased vegetation growth can increase the natural rate of C input and thus potentially SOC stocks (Ryals et al. 2013)."*

*Malik MA, Khan KS, Marschner P, Fayyaz-ul-Hassan. 2013. Microbial biomass, nutrient availability and nutrient uptake by wheat in two soils with organic amendments. Journal of Soil Science and Plant Nutrition 13(4):955−66. doi: 10.4067/s0718-95162013005000075*

L69: Please use references correctly, (e.g. Hirte et al. 2021 looked at root biomass and root-shoot ratio, no crop yields!)

*We agree that the current use of that reference is confusing, and propose to change the formulation to "and of roots in farming systems (Hirte et al. 2021)"*

L78-79: This statement is completely out of context.

*We propose to change this to: "This variability derives partly from the variable physical properties of soil, but can also depend on land use history or on small- and large-scale topography (Wang et al. 2020), and highlights the need for more field-based data collections—in particular under experimental conditions that combine soil amendments and drought."*

L81-83: Too much info for one sentence. Split into two.

*We propose to change this to: "Here, we present the results of a field experiment designed to assess how soil C amendment and reduced precipitation affect both soil and vegetation C pools. The changes were observed at various soil depths, in different plant organs, in two grasslands, and at two catenary positions."*

L100: What are "control" plots. You need to introduce this term before using it the first time.

*We propose to introduce the term as "ambient precipitation, no compost treatment".*

L105ff: Provide coordinates and masl for every location.

*This data will be added in the supplement, as well as soil texture.*

L111-112: Provide exact textural composition for every location.

*Will be added (see above)*

L180-181: Sentence is unclear. Please split up into two sentences.

*We propose: "The effect of the treatments was tested on all plots from the 2022 dataset. Values for root biomass and root traits were log-transformed first."*

L198: To which time period does this refer to? Please provide information.

*This refers to growing season averages (April-October), mentioned in the methods section but can be added here.*

L198-202: Ok, there was no significant difference but I would like to see the actual numbers. Can you summarize soil moisture data in a table or display it in a figure? This is important information since drought (and thus soil moisture) is key to the presented study.

*We agree with this comment and propose to add a figure summarizing soil moisture data in the supplements.*

L203-204: Ditto, please provide these numbers (pH, nutrients) in one form or another.

*Will be added (see above)*

L215 (and many other occasions): What does the number after the +/- indicate? Standard deviation, confidence interval, standard error? These needs to be indicated.

*The numbers indicate the standard deviation, this will be added*

Fig. 2 (and elsewhere): You cannot provide C in % since this is ambiguous and not an SI unit. It needs to be presented as a mass ratio!

*As per response to another reviewer, we propose that the current data in figure 4 may be complemented with an analysis of soil mass change over time. We can also report soil organic C contents in mg C per g soil mass.*

L299: In which figure/table can I find this information. In general, please refer to figures and/or tables in the discussion. It is important to know which of your data supports finding X and Y.

*The statement is supported by Fig. 1, and this information will be added to the text.*

L291-306: This entire paragraph contains 2 (!!!!) references but there is a huge body of literature available on effects of organic amendments on SOC, not only in grasslands but also croplands. Hence, you need to acknowledge some of the work done in this area and 2 references just don't do the job here.

*We propose to revise this paragraph as suggested by reviewer #1, which includes adding appropriate references here and in the introduction where they can help support our motivations for the design of the experiment and hypotheses, as suggested in the previous comments. There are several studies that report increased SOC after < 4 years (see Gravuer et al. 2019), but also long-term studies like Ryals et al. (2015). Multiple references were added to the paragraph.*

*Gravuer K, Gennet S, Throop HL. Organic amendment additions to rangelands: A meta-analysis of multiple ecosystem outcomes. Glob Change Biol. 2019; 25: 1152–1170. https://doi.org/10.1111/gcb.14535*

*Ryals, R., Hartman, M.D., Parton, W.J., DeLonge, M.S. and Silver, W.L. (2015), Long-term climate change mitigation potential with organic matter management on grasslands. Ecological Applications, 25: 531-545. https://doi.org/10.1890/13-2126.1*

L307-316: Same here, there is a plethora of work done on root responses (and C allocation responses within plants) to changes in nutrient availability. I strongly recommend that you check such literature. Not only will credit be given where credit is due, but careful evaluation of relevant plant (eco-)physiological literature will improve the quality of the manuscript as a whole. E.g. this very paragraph needs to put the results obtained here into a plant eco-physiological context (e.g. what does higher tissue density and the other root traits that were evaluated mean with respect to C allocation, soil exploration potential, etc?).

*We agree with this comment, and propose to elaborate the content of the paragraph with reference to Cleland et al. (2017; lower biomass allocation to roots after increased nutrient*

*availability), Bardgett et al. (2014; higher root tissue density as correlated with resource-conservative acquisition strategies). The paragraph was partly re-written.*

*Cleland, E.; Lind, E.; DeCrappeo, N.; DeLorenze, E.; Wilkins, R.; Adler, P., et al. (2019). Belowground Biomass Response to Nutrient Enrichment Depends on Light Limitation Across Globally Distributed Grasslands. Ecosystems, 22(7), 1466-1477. http://dx.doi.org/10.1007/s10021-019-00350-4*

*Bardgett, R.D., Mommer, L. & De Vries, F.T. (2014) Going underground: root traits as drivers of ecosystem processes. Trends in Ecology & Evolution, 29, 692–699.*

L318: Why did the experimental set up not allow for this? You could have sampled soil and measured e.g. respiration to assess microbial activity. Just because it has not been done does not mean that it was not possible. Please don't get me wrong, it is completely ok that microbial activity was not assessed (it is impossible to measure everything) but the reasoning you provide here ("the set up did not allow for it") does not seem to be honest to me.

*We agree that the formulation is inappropriate, and suggest changing to "Microbial activity and microbial biomass can be higher afteras a result of compost addition (Sarker et al. 2022; Gravuer et al. 2019). Here, the limited effects…"*

L352-354: Do you have any data on that, i.e. the composition and its response to time/treatment/location? I think this should be included in the paper.

*This data is reported in another study, currently a manuscript. To avoid confusion, we propose to modify the sentence to: "Roots in particular where not significantly affected by drought (Fig. 2). Because we do not know which plant species the sampled roots belong t, we cannot…"*

L359-361: Similar as the comment above: How? What are the connections between root traits and environmental conditions? How can these responses of plants to drought be interpreted?

*We propose to discuss this with reference to studies mentioning climate as a strong predictor of root trait variation (Freschet et al. 2017), higher root tissue density as correlated with resource-conservative acquisition strategies (Bardgett et al. 2014) and longer root life span (Ryser, 1996)*

*Freschet GT, Valverde-Barrantes OJ, Tucker CM, Craine JM, McCormack ML, Violle C, Fort F, Blackwood CB, Urban-Mead KR, Iversen CM, Bonis A, Comas LH, Cornelissen JHC, Dong M, Guo D, Hobbie SE, Holdaway RJ, Kembel SW, Makita N, Onipchenko VG, Picon-Cochard C, Reich PB, Riva EG, Smith SW, Soudzilovskaia NA, Tjoelker MG, Wardle DA, Roumet C. 2017. Climate, soil and plant functional types as drivers of global fine-root trait variation. Journal of Ecology 105:1182–1196. doi:10.1111/1365-2745.12769.*

*Bardgett, R.D., Mommer, L. & De Vries, F.T. (2014) Going underground: root traits as drivers of ecosystem processes. Trends in Ecology & Evolution, 29, 692–699.*

*Ryser, P. (1996) The importance of tissue density for growth and life span of leaves and roots: a comparison of five ecologically contrasting grasses. Functional Ecology, 10, 717–723.*

L393-396: Organic amendments cannot "increase soil moisture" as you write. What they can do is to increase water infiltration and retention capacity, which in turn may lead to more water stored in soil. Please be specific on that. Moreover, would you expect that such an effect is detectable after such a short period of time, i.e. the three years covered in the presented study? Maybe these effects would appear later, e.g. after 10 years of continuous organic amendments. This should be discussed (as I mentioned in my general comment 4).

*The formulation was unclear. We propose to change to "While previous studies indicate increased soil water retention after soil amendments (Franco-Andreu et al. 2017; Ali et al. 2017), in our study compost-treated drought plots did not have higher soil moisture than the untreated drought plots 3 years after compost application, which leads us to reject our third hypothesis".*

*The time aspect can also be included to nuance the conclusion*

L414-416: Yes, I agree 100%! This is a very nice conclusion of the story. In fact, that very sentence could be a guidance for the revisions.

*Thank you, we will consider this when applying all recommended changes to introduction and discussion.*

**Reviewer #3**

*Dear anonymous reviewer,*

*We greatly appreciate the constructive feedback provided on our manuscript. Should we have the opportunity to submit a revised version, we are confident that we can address all concerns raised and significantly enhance the quality of our manuscript. Attached below is a detailed plan outlining how we propose to address each comment. Kind regards on behalf of all listed authors,*

*Daniela Guasconi*

**General comments**

You studied the effect of compost amendments in an artificial drought treatment and looked into the effect on C allocation and belowground/aboveground biomass. You show that effects

are overall most pronounced in the topsoil and that location and climate are causing overlaying effects, often stronger than your compost treatment.

Relevant research but the overall story line should be improved. Make sure that the reader has a strong guidance on what you are doing and why you choose to do it this way. Sometimes you use quite long and confusing sentences, try to be more concise. Especially the discussion can be improved and all the presented results should be incorporated (and some of the not shown results could be interesting additions). In some parts citing more recent publications could help to give your publication more context, be careful with overgeneralizations from literature. Make sure that your conclusions are always logically derived from the previous paragraph. A positive control with comparable nutrient in synthetic form could have improved confidence of your findings. Right now, your draft has some major shortcoming, but with the incorporation of the below mentioned points and the feedback of the other reviewers, this can turn into a decent and relevant publication.

*We thank the reviewer for their comments. In our responses below, we outline our plan to address their criticism.*

**Does the paper address relevant scientific questions within the scope of SOIL?**
Yes, sustainable fertilizer and response to climate extremes via adaptation and mitigation. What is the impact of drought on aboveground/belowground biomass and can compost mitigate this effect? What is the mean residence time of compost C in soil under simulated futuristic scenarios?

**Does the paper present novel concepts, ideas, tools, or data?**
Yes, they generated data of combined drought and organic amendments with a special component. It is novel, that also sufficient attention is paid to the temporal and spatial aspect, in contrast to many studies that only include time or location.

**Does the paper address soils within a multidisciplinary context?**
Yes, climate, soil, aboveground and belowground interactions relevant for scientists, policymakers and farmers.

**Is the paper of broad international interest?**
Partly, experiments are conducted in Sweden but relevant for similar soils and climate.

**Are clear objectives and/or hypotheses put forward?**
Yes.

*Comments up to this point are mostly confirming that our work is relevant and suitable for the journal SOIL, but do not require specific changes.*

**Are the scientific methods valid and clearly outlined to be reproduced?**
Yes partly, descriptions of methods and protocols should be more detailed.

The experiment could draw stronger conclusion when a positive control with mineral fertilizer would have been included. Overall, be more cautious to provide sufficient information on methodology, details matter. Also complete protocols could be included in supplementary information.

*Additional information on our sampling and analysis protocols can be easily added in a revised manuscript. Reviewer #2 had a similar comment regarding the analysis of root data. We agree that a control with mineral nutrient addition could have been informative, but our original question was (and still is) about the consequences of compost amendments on soil organic carbon (either directly or indirectly through vegetation feedbacks) and whether such effects change under drought.*

**Is the soil type/classification adequately described?**
No, data on soil composition is missing.

*This was an oversight on our part. Soil properties are described in a previous publication on the same sites (Guasconi et al. 2023, Vegetation, topography, and soil depth drive microbial community structure in two Swedish grasslands, FEMS Microbiology Ecology 99(8), fiad080, https://doi.org/10.1093/femsec/fiad080), but we can add a table in the supplementary materials summarizing soil properties (texture, nutrients, pH).*

**Are analyses and assumptions valid?**
Yes, mostly. See comments below.

**Are the presented results sufficient to support the interpretations and associated discussion?**
Yes, mostly. See comments below.

**Is the discussion relevant and backed up?**
Yes, mostly. Some more references could help to see your research in a broader context. Also make sure that statements are combined in coherent paragraphs.

**Are accurate conclusions reached based on the presented results and discussion?**
Yes.

**Do the authors give proper credit to related and relevant work and clearly indicate their own original contribution?**
Yes.

*The above comments do not require a specific response.*

**Does the title clearly reflect the contents of the paper and is it informative?**
No, the title is not a good match with context. The focus and experimental design cannot support this title. See below.

*We propose a new title: "Experimental drought and soil amendments affect grassland above- and belowground vegetation but not soil carbon stocks"*

**Does the abstract provide a concise and complete summary, including quantitative results?**
No, overall quantitative statements are missing. How much compost was applied e.g., t/ha, kg N / ha? What soil layers are you referring to? What were the absolute changes? What about the overarching natural climate variabilities you identified?

*Abstracts need to be concise, so it is difficult to strike a balance between details and length. We would prefer to avoid reporting quantitative information on the treatment levels, which are not essential when reading the abstract, but we can be more specific about soil depths. We also find trends and correlations in our results more interesting than absolute changes.*

**Is the overall presentation well structured?**
Yes, partly. Discussion could be more focused.

**Is the paper written concisely and to the point?**
Yes, mostly. Some paragraphs are repetitive. Also make sure to use one term for one concept throughout the paper to prevent ambiguity. Choose the more common wording when possible. (see comments below)

*We can revise the text keeping into account this comment and double-checking to avoid repetitions and to use consistent terminology.*

**Is the language fluent, precise, and grammatically correct?**
Yes.

**Are the figures and tables useful and all necessary?**
Yes.
**Are mathematical formulae, symbols, abbreviations, and units correctly defined and used according to the author guidelines?**
Yes.

**Should any parts of the paper (text, formulae, figures, tables) be clarified, reduced, combined, or eliminated?**
Yes, fig. 3 and 4 could be completed by adding the data from the compost treatments.

**Are the number and quality of references appropriate?**
Yes, but more and recent references could be beneficial. And make sure they support your arguments, see below.

*The above comments do not require a specific response, except the suggestion about the figures (see details in our responses below).*

**Is the amount and quality of supplementary material appropriate and of added value?**
More details should be provided. E.g., vegetation analysis, detailed description of used methods and protocols. Soil moisture, volumetric soil water content, soil nutrient content...

*We can add more details in a revised manuscript.*

**Specific comments**

**Title**

It is misleading, you need to emphasize in your story more on the temporal aspect and show all the data from different years or change the title. The development in the years 2020 and 2021 would have added significant resolution to the temporal dynamic. Now you have two time points and mostly know that most C is lost in one way or another. Spatial variability is a side endeavor and should not be part of the title in this version

Here are some key words I suggest.

Field experiment + grassland + aboveground +belowground + 13C + compost + Sweden

*We suggest to change the title from "Spatial and temporal variability in soil and vegetation carbon dynamics under experimental drought and soil amendments" to the more focused "Experimental drought and soil amendments affect grassland above- and belowground vegetation but not soil carbon stocks"*

**Abstract**

L 8: Soils are the biggest terrestrial C pool, but not the biggest C pool on this planet, sequestration potential is not equal to C stocks.

*We agree with this correction, we suggest changing to "largest terrestrial carbon (C) pool"*

L 8-12: I would keep this for the introduction and come straight to the point of your own research.

*We suggest reducing and synthetizing the information contained in the first two sentences.*

L 14-15: I would like to see some numbers here, also important to note with soil layer you are referring to. How long is your drought and how severe? What soil and what climate? All the information someone would need to consider if this research is relevant for them.

*We would like to keep the abstract short and concise consistently with the publication guidelines, and therefore not include too many numbers or specifics. However, we will consider adding a few more details about the setting of the experiment (eg. that the study was carried out in two Swedish grasslands, and that the drought was a partial precipitation reduction extended over four growing seasons).*

L 17: You only have information about biomass, C allocation is a hypothesis.

*We agree, this will be corrected to biomass*

L 18-19: Later you state that you cannot exclude strong overlaying legacy effects of climate and precipitation extremes in the year before the study and between your sampling points. That's a fair limitation and important to know.

*This can be included in the abstract*

**Introduction**

You must make clear that there are several organic soil amendments and that you focus on your study for good reasons on compost. And from there on focus only on compost.

L 24: Better can act as C sink (under specific conditions)

*This was added*

L 29: Just manure, there are also different forms of compost and biochar.

*This was corrected*

L 33-34: This is a very controversial point in the soil science community, how big could the potential contribution to climate change mitigation be? Concepts like C saturation and the non-permanent character of organic soil amendments are important to consider.

*We think it's fair to state that this kind of treatments has been proposed and is currently being discussed, and perhaps we could add more references in testimony of the ongoing scientific debate. While our study does not focus on C saturation, we want to make the point that field studies such as this one can provide valuable empirical data to the debate. The debate on saturation was mentioned in lines 340-341*

L 35-36: You mention biochar above…the aim is to bypass the vegetation and directly store recalcitrate C rich material in the soil (POM and MAOM). There is also a significant contribution of necromass to soil C.

*We propose to reformulate this sentence as per suggestions by reviewer #2, and may include these points. "Consequently, soil C management via compost amendments aims at transferring plant-derived organic matter to facilitate C accumulation in the soil C pool in specific locations, where it can be retained over long time scales".*

L 36: "over long time scales", very vague this is one of the most relevant questions and need some more attention.

*We refer here to stabilized C that can remain in the soil longer than timescale relevant for climate change—i.e., decades to centuries. We propose to nuance this statement by clarifying this and adding appropriate references, such as Shi et al. (2020).*

*Shi, Z., Allison, S.D., He, Y. et al. The age distribution of global soil carbon inferred from radiocarbon measurements. Nat. Geosci. 13, 555–559 (2020). https://doi.org/10.1038/s41561-020-0596-z*

L 37-41: confusing sentence, but relevant thought. Try to make the separation between direct and indirect effects clearer. Primer and priming are here referring to separate concepts, maybe use words that have less potential for ambiguity. Also, some more references would strengthen this part.

*We agree with the suggestion, and propose changing to: "C amendments add C to the soil in two ways: directly, by moving plant biomass from one location to another, and indirectly, by promoting plant growth (Ryals et al. 2016). Compost is rich in organic matter, which serves as a substrate for soil microorganisms. As microbes decompose this organic matter, they release nutrients in forms that plants can readily absorb (Malik et al. 2013). In turn, this increased vegetation growth can increase the natural rate of C input and thus potentially SOC stocks (Ryals et al. 2013)"*

*Ryals, R., V. T. Eviner, C. Stein, K. N. Suding, and W. L. Silver. 2016. Grassland compost amendments increase plant production without changing plant communities. Ecosphere 7 (3):e01270. 10.1002/ecs2.1270*

*Ryals R, Silver WL. Effects of organic matter amendments on net primary productivity and greenhouse gas emissions in annual grasslands. Ecol Appl. 2013 Jan;23(1):46-59. doi: 10.1890/12-0620.1. PMID: 23495635.*

L 43: What's is the difference between roots and belowground plant organs in this context?

*As currently formulated the sentence is confusing, and we suggest removing "belowground plant organs"*

L 54-55: root biomass is also affected by the crops and they are mostly annual crops in contrast to perennial grasslands.

*We propose specifying that much of the difference is due to crops being annuals*

L 59: (Luo et al. 2018) needs to be used with caution……organic amendments are including treatments that combine mineral fertilizer + organic amendment and not all organic amendments show a positive effect size. And besides that, this study is focusing on crop production.

*We did write that the study focuses on crop yields, to indicate that amendments in general have shown positive effects when applied to crops. However, if it's a source of confusion we propose to remove the reference and the mention of crop yields.*

L 61: (Fischer et al. 2019) is only looking on biochar, you cannot generalize to all organic amendments you do research on compost. I cannot check all references but make sure to be sincere.

*We agree that the formulation is ambiguous, and suggest to specify that the study refers to biochar as to avoid confusion.*

L 63-64: Please elaborate the expected effects they are very important for your discussion.

*This point can be expanded. There are multiple mechanisms at play that can alter the soil carbon balance under drier conditions. We propose to elaborate and explicitly mention the mostly negative effects of drought on vegetation growth above- and belowground (Guasconi et al. 2023) and on plant C allocation (Hasibeder et al. 2015). C inputs tend to be lower, but so are C losses due to reduced microbial access to soil organic C in dry soils (in addition to physiological impairment) (Moyano et al. 2013 ; Fuchslueger et al. 2019). The effect of decreased input is generally stronger, so that soil C stocks tend to decrease under experimental drought (Deng et al. 2021).*

*Guasconi, D., Manzoni, S., Hugelius, G. 2023. Climate-dependent responses of root and shoot biomass to drought duration and intensity in grasslands–a meta-analysis, Science of The Total Environment, Volume 903, 166209, ISSN 0048-9697, https://doi.org/10.1016/j.scitotenv.2023.166209.*

*Deng L., Peng C., Kim D.-G., Li J., Liu Y., Hai X., Liu Q., Huang C., Shangguan Z., Kuzyakov Y., Drought effects on soil carbon and nitrogen dynamics in global natural ecosystems, Earth-Science Reviews 214 (2021), 103501. https://doi.org/10.1016/j.earscirev.2020.103501.*

*Hasibeder, R., Fuchslueger, L., Richter, A. and Bahn, M. (2015), Summer drought alters carbon allocation to roots and root respiration in mountain grassland. New Phytol, 205: 1117-1127. https://doi.org/10.1111/nph.13146*

L 71-72: This statement needs some explanation/clarification.

*We propose changing to "Above- and belowground biomass may respond differently to soil amendments (Garbowski et al. 2020), which is expected since roots and shoots…"*

L 81: The effect of compost.

*This can be corrected*

L 87-90: Belongs to the material and methods part.

*Can be moved to methods*

L 91: What is the expected mechanism for increased soil C content and plant growth?

*We expect that adding an organic amendment has a direct effect on soil C (due to increasing C inputs in the system), but also an indirect effect by promoting plant growth. This indirect effect would occur thanks to nutrient release from the added compost and improved soil properties (lower bulk density).*

L 94-95: Would be interesting to state the expected effect on native soil C and compost C.

*The second hypothesis refers to drought alone, not in interaction with compost C, which is addressed in our third hypothesis. Therefore, we would not add a more specific expectation on the consequences on native vs. added C.*

L 94-95: Weak positive or weak negative effect on SOC % or SOC stocks?

*We expect a weak negative effect. We are referring to stocks, as the reasoning is based on a mass balance (change = input – output). This can be specified in the revision.*

L 96-97: Any expectations on aboveground and belowground interactions?

*We prefer to delimit this specific hypothesis to changes in soil moisture.*

L 98-102: This is part of the introduction and should be moved in before the hypothesis, eventually you can derive a 4th hypothesis for the spatial aspect. Also, you do not mention any temporal dynamic in your hypothesis.

*This part can be moved before the hypotheses*

**Methods**

L 106-107: please provide exact positions of fields with coordinates and mean annual precipitation ant T and climate…

*This information can be added*

L 109: grazing and haymaking need to be elaborated a bit more, how often are the fields cut/grassed (what animals), how much aboveground biomass is harvested, what is the nutrient composition, with plant species grow there? What is the management history? C content?

*All this information refers to the land use history (retrieved from local farmers), since there was no management during the duration of the experiment. We will add information relative to grazing (cows) and refer to another publication where more details of vegetation are provided (Roth et al. 2023)*

*Roth, N., Kimberley, A., Guasconi, D., Hugelius, G., & Cousins, S. A. O. (2023). Floral resources in Swedish grasslands remain relatively stable under an experimental drought and are enhanced by soil amendments if regularly mown. Ecological Solutions and Evidence, 4, e12231. https://doi.org/10.1002/2688-8319.12231*

L 111: Please provide exact composition of soil (e.g., % clay etc.) and the location of the plots in the field (e.g., in the middle or at the edge…)

*This information will be added to the supplements*

L 112: I would like to see a short description of the composting process details, compost pile size, composting time, nutrient analysis at the end… storage time…

*We suggest adding the more details to the composting description such as: "The compost was made of Zea mays with a C:N ratio of 9.8 and $\delta 13$ C value of about -15.39‰. After the seasonal corn harvest (summer 2019) the green parts of the plants were collected in an open field. The piled material was regularly stirred to promote the composting process, and the resulting compost was collected and applied in mid-February 2020 as a thin surface layer of ca. 11 kg per m2 (wet weight), similar to the procedure described in Ryals and Silver (2013)..". Nutrient analyses will be mentioned together with the soil nutrient analyses.*

L 113: 11 kg / m2 = 110 t / ha = ~ 5+ times the "normal" application rate in agriculture please elaborate why you choose this high amount. Please provide dry matter content of compost.

*The compost application was designed after Ryals and Siler (2013), this will be specified.*

L 115: 11 kg / m2 ~ 5 kg dry matter / m2 à 0.54 kg C / m2 à ~10 % of C (which is quite low for compost), estimated, you did not measure it?

*While we did measure the C content of the compost (as %) and the water content, it is difficult to get a confident value for the exact amount of C added to the plot. This is because of the error margin given by field conditions (The compost was added 10 kg at a time and spread out across the plot surface, but the compost bags might have differed in weight by a few*

*hundred grams). Perhaps "estimated" is a misleading term, we suggest changing to "average"*

L 121-122: Why is the treatment different in the first year?

*This is because the research project started in spring 2019 and it took some time to build the rainout shelters and set everything into place.*

L 135: for 3 h? Please provide all relevant details.

*We will provide the time details of the process*

L 131-138: Please give more elaboration on the sampling procedure.

*Some comments on this were given also by another reviewer, and we propose to modify this section as follows: "Samples for soil bulk density were collected with a large fixed volume root auger with a sharpened cutting edge (8 cm diameter and 15 cm in length; Eijkelkamp, The Netherlands). Three 15 cm segments were collected sequentially using the same hole, reaching a total depth of 45 cm. Upon extraction, the cores were cut into 5 cm segments, and the bulk density was determined after drying the samples at 105 °C…". We are unsure however about other specific information that should be added on this (except what was already mentioned in the revision comments).*

L 148-149: What is the reasoning for the different sampling depth'?

*As per response to reviewer #1, the roots in our study system are very shallow, and we estimated that 95% of the total root biomass in the plots was < 30 cm. The amount of roots in the deep samples (30 – 45 cm) was so low that we considered those samples to be a control for maximum rooting depth. Considering the difficulty of the sampling in the clay-rich soil, especially in the late summer when root samples were collected, we chose one plot per treatment and location for deep root sampling. This can be specified in the text.*

**Results**

L 198: 16 %, what are the absolute numbers?

*We added a figure summarizing this data in the supplements*

L206-209: How much compost derived C was in the deeper soil layers you sampled up to 1 m? How much of total compost C could be found back? Consider making 2 separate sentences for the 2 treatments.

*We suggest clarifying these results by including both treatments (drought, drought + compost) in the figure as per suggestion from another reviewer, and making two separate sentences as suggested here*

L 213-230: Are you comparing 2019 with 2022 or the effect of the different treatments in 2022? Both would be interesting.

*Only the effects of the treatments in 2022 are observed here, as mentioned in the methods (L 180)*

L 215: 23 % is the average of the 3 years or in 2022? Would be also interesting to see the effect on the aboveground biomass over time, since you only applied compost once and harvested every year….is the effect on the aboveground biomass fading over time?

*Only the effects of the treatments in 2022 are considered here, as mentioned in the methods (L 180). The effects of the treatments on aboveground biomass over time are presented by Roth (2023) and are outside the scope of this manuscript. The choice of focusing on year 2022 is motivated by the need to conduct consistent analyses of all the data, and we do not have yearly data for soil C and roots.*

L 223: 4 % is significant? Please provide p value. See above, effect in different years maybe the natural precipitation variation had also effects on aboveground biomass harvest…

*This is unclear and will be corrected as per suggestion from reviewer #1*

L 223-224: still part of the previous paragraph

*This can be corrected*

L 225: Start of new paragraph, but could also be part of discussion…

*We would follow the suggestion to start a new paragraph*

Figure 2: Would also be interesting to see the differences compared to 2019…temporal dynamics

*Our preference was to presenting the treatment effects on the 2022 dataset only, because of possible confusion created by the temporal changes on the 2019-2022 comparison*

L 250-252: This conclusion does not align with the rest of the paragraph and could be as well considered for the discussion part.

*This could be moved to the discussion*

L 259: You mention land-use history and vegetation composition but never show any data? Same for catenary position, what is the exact difference?

*This is perhaps causing some confusion, which hopefully will be avoided when adding more details about the locations in the methods section. This will include how the two grasslands differ in their relative abundance of grasses, forbs and legumes (table added in the supplements). The effects of grassland and catenary position on aboveground biomass are illustrated in fig. 3, we will add a reference to it.*

Figure 3: Why are you not showing the site specific difference of the treatments?

*We suggest adding a figure illustrating the interactions between site and treatments in the supplements*

L 270: Was this effect also evident in the compost treatments? And had the year or the compost a stronger effect?

*We reported effect sizes (Table 1) to allow for comparison between year, location and treatment. As mentioned earlier, we chose to focus the treatment effects only on the 2022 dataset*

L 275: what is the confidence interval for 8.4 %?

*This can be added*

L 274-277: That's very hard to follow, consider to make 2 sentences. Also, how can be the root biomass less in a thicker layer (top 5 cm compared to top 15 cm)?

*The sentence can be restructured. The numbers reported regarding the root biomass in the top 15cm are an error that will be corrected, we thank you for identifying it. This does not affect the other calculations.*

Figure 4: What happened in the other year I want to see temporal dynamics, even when the difference is not significant

*We are not sure if "other year" refers to "other years" (2020, 2021). If so, the soil variables were not measured during those years as we did not expect to see yearly differences and because of practical limitations in the sampling.*

**Discussion**

L 293-294: Would be helpful to state the measured difference, how can you not know the exact added C per m2 if you know the C/N of your compost?

*As per previous comment, perhaps the term "estimated" is misleading and might be changed to "average"*

L 294-297: That's not correct and an oversimplification of Sarker 2022, compost has a direct effect after application on many soil processes. Compost is not inert, compost is decomposing.

*We agree that the current formulation is inappropriate and confusing, we meant to say that the effects can be potentially persistent over several years, not that short-term effects cannot be expected. We propose to reformulate and clarify that our reasoning refers to the timescales considered, i.e. that we might expect to detect an effect of the compost amendment also longer time after the treatment. ("Despite evidence that compost amendments can lead to SOC accumulation already within two years after application (Gravuer et al. 2023), it is likely that the effect of our treatment on soil properties and soil C will persist beyond the 2022 sampling.") (L 323-326)*

L 298-300: This conclusion is contradicting the previous statement that compost is recalcitrate and no short-term effects can be expected. You still lost 80 + % of your initial C over 3 seasons…so we can expect that the composition of these 80 % had an impact on the soil, plants and microbes. Amlinger et.al., 2003 (doi:10.1016/S1164-5563(03)00026-8) could be insightful.

*We agree, and as per previous comment, we hope that the rephrasing will clarify the text*

L 300-302: Your control plots (Figure 4) show also significant increases in aboveground biomass in 2019 compared to 2022. This limits the causality of the compost as explanatory variable.

*Aboveground biomass increased between 2019 and 2022 (Figure 4). This was noted also by reviewer #1. Total soil N was measured (line 155) and reported (line 250). We propose to reformulate the sentence as "the significant increase in aboveground biomass three years after the compost application could partly be explained by the persistence of favorable plant growing conditions, such as increased N in the soil. This mechanism was invoked by Oladeji et al (2020), and may interact with precipitation-related interannual variability in plant growth (Sala et al. 2012)"*

*Sala Osvaldo E., Gherardi Laureano A., Reichmann Lara, Jobbágy Esteban and Peters Debra 2012 Legacies of precipitation fluctuations on primary production: theory and data synthesisPhil. Trans. R. Soc. B367: 3135–3144.*

*Oladeji, O., Tian, G., Lindo, P., Kumar, K., Cox, A., Hundal, L., Zhang, H. and Podczerwinski, E. (2020), Nitrogen release and plant available nitrogen of composted and un-composted biosolids. Water Environ Res, 92: 631-640. https://doi.org/10.1002/wer.1260*

L 302-305: Did you measure the growing season? That's not a supporting reference; you must be more cautious and read the references you are using carefully.

*The sentence as currently formulated is confusing, we suggest changing to: "our results suggest that compost treatments might benefit the ecosystem C balance indirectly through increased biomass production, such as in this case, or as an extended growing season, such as in Fenster et al. (2023).*

L 308: Why should a plant invest in belowground biomass when sufficient nutrients are available in proximity. A tradeoff would suggest that the plant is sacrificing something, and that's not the case. A trade-off implies either or; what we see here is possibly an adaptation to nutrient supply.

*We agree with this comment, and trade-off might be inappropriate in this case. We suggest changing to "..hypothesis, possibly in response to the increased nutrient supply"*

L 313-314: You can only draw conclusions on the biomass allocation.

*We agree, this can be changed*

L 314-316: Consider that the roots grow preferably where the nutrients are accumulated.

*This can be added in the text*

L 321-322: Are there references that observed similar findings/effects? Compost gets decomposed over time (as also evident from your 13C data), maybe annual compost amendment would work better. Also, your compost application could induce priming of native soil C.

*We agree that annual compost addition would have a stronger effect, but our initial goal was to assess if the positive effect of a single addition observed in the Marin Carbon Project (Ryals et al. 2013; Ryals et al. 2015) could be replicated in other grasslands and under drought.*

*Regarding similar findings, we propose to refer to Borken et al. (2002).*

*Ryals R, Silver WL. Effects of organic matter amendments on net primary productivity and greenhouse gas emissions in annual grasslands. Ecol Appl. 2013 Jan;23(1):46-59. doi: 10.1890/12-0620.1. PMID: 23495635.*

*Ryals, R., Hartman, M. D., Parton, W. J., DeLonge, M. S., & Silver, W. L. (2015). Long-term climate change mitigation potential with organic matter management on grasslands. Ecological Applications, 25(2), 531–545.*

*W. Borken, A. Muhs, F. Beese. Application of compost in spruce forests: effects on soil respiration, basal respiration and microbial biomass. Ecol. Manag., 159 (2002), pp. 49-58*

L 327-329: Again, I want to see this data that seems to explain a lot.

*We propose to add in the supplements a table showing the relative abundance of grasses, forbs and legumes in 2019 (as percentage of total dry weight of live plant biomass excluding mosses, average values per site, n = 6):*

| site | % grasses (n = 6) | % forbs (n = 6) | % legumes (n = 6) | Average biomass increase (g m$^2$) 2019-2022 (compost-treated plots, n = 3) |
|---|---|---|---|---|
| Tovetorp High | 66,4 | 23,9 | 9,7 | 420,3 |
| Tovetorp Low | 88,4 | 10,6 | 1,0 | 581,7 |
| Ämtvik High | 66,0 | 26,2 | 7,8 | 234,7 |
| Ämtvik Low | 54,8 | 34,6 | 10,5 | 193,7 |

L 331-332: Tradeoff is implying that one trait is enhanced on cost of another trait. The trade off could be a higher investment in (deeper) roots to find water in drought conditions and consequently less investment in aboveground biomass. Try to make your argumentation more logical.

*We agree with this comment, and that "tradeoff" might not be the most appropriate term in this context. We suggest reformulating to: "..indicating preferential biomass allocation and resource investment to belowground organs under precipitation reduction"*

L 337-348: I would make a new paragraph in you discussion about underlaying natural variation in precipitation and possible implications for your results, otherwise it can be confusing to follow when you are referring to your drought treatment and when to the natural drought.

*Our thinking was that this information should be mentioned in order to discuss its possible implications on the results, but that this should be delimited since the conclusions we can draw from this hypothesis are limited. After suggestions from another reviewer, details of the variability in precipitation were moved to the methods.*

L 353: could be relevant data for site description…

*We suggest to include the data in the supplements*

L 382: No, your data shows that aboveground biomass was not significantly different from the control treatment.

*This was pointed out by another reviewer and will be corrected*

L 383-384: Any reference?

*The paragraph was restructured and the sentence removed.*

L 385-386: Please show the data. Why is there little compost C in 10-25 cm and more in 20-45 cm?

*This will be shown when remaking the figure. We cannot however offer a good explanation for the data*

L 386-388: You cannot make absolute statements without supporting data.

*We agree with this observation; the current formulation is inappropriate. We suggest nuancing to: "This would imply that any soil C emission pulses at rewetting were not sufficient to compensate for the possibly lowered microbial activity during the soil moisture dry-downs."*

L 393-395: Why are you not showing the data for soil moisture?

*This result is presented in line 231-232 and we did not deem it necessary to include more graphs or data tables, but we added a figure in the supplements.*

L 400-402: Why would a shift from aboveground to belowground C allocation lead to more evapotranspiration? Consider rephrasing this sentence to make it easier to follow.

*We propose to rephrase as "This improved capacity for water absorption could potentially offset any compost-induced increase in soil water retention capacity"*

**Conclusion**

L 409-410: You have no information on plant and root C.

*We propose to reformulate this as: "Compost amendment and drought had distinct effects on plant shoot and root growth"*

L 412: What are the potential contributions to ecosystem C modelling?

*This sentence will be removed, as per response to reviewer #1*

L 414: We need in all the previous parts of this text information and data on topography and plant community composition.

*This will be referred to as appropriate in the text and in the supplements*

L 415: What biotic and abiotic factors played an important role in your study?

*We suggest modifying the sentence as follows: "This suggests that ecosystem C dynamics are better understood by considering multiple factors including climate, environmental history and biotic interactions, and which…"*

**Technical corrections**

Make sure that it is clear for the reader where you talk about trends in your data and when these trends are significant results.

Try to be consistent with your wording throughout the text and consider using more common words. One word per concept is sufficient and leads to less ambiguity.

L 11-13: Confusing sentence, consider making two clear sentences.

*As mentioned in response to reviewer #2, we propose to change the sentence to: "However, soil C dynamics are deeply linked to vegetation response to changes in both management and climate. The responses to environmental change may also be manifested differently in roots and shoots."*

L 13: Field experiment is important to mention.

*This will be added*

L 31-32: human activities very vague term, agricultural activities may better suited.

*We agree with this change*

L 83: "catenary positions" this is the first time I read this term and I don't find a clear definition in your context on the internet. Please consider alternative wording.

*'Catenary position' is a common term referring to the location of sites along a 'catena'—i.e., a series of soils along a slope. We would keep the term, but we can explain it at the first mention.*

L 65: soil capacity to retain moisture = water holding capacity?

*We agree with this change*

L 136: "very low" = low, no need for an opinion and also no comparison provided.

*Noted*

L 157: thickness = density?

*The formulation is confusing, we propose changing to "and soil C stocks (expressed as kg/m$^3$) normalized by soil sample thickness"*

L 270: "large changes" is a subjective description and needs some context, maybe let the numbers speak for themselves.

*We can remove "large"*

Soil C amendment = (organic) soil amendments often abbreviation OA

Rangelands = grasslands?

*These last two points will be revised in the text*

---

## Referee Report (RR1)

**Abstract:**

14: Compared to unfertilized controls or mineral fertilized control be specific.

**Introduction:**

25-26: Transfer C to the soil via root exudates and decaying aboveground and belowground organic matter.

29: Consider introducing the concept 4/1000 to the reader.

29-30: Mitigating anthropogenic GHG emissions by restoring/enhancing soil C stocks.

31-32: It's not only about countering tillage, it's about minimizing all agricultural management strategies that could have negative impacts on SOC stocks. i.e., no/minimal-tillage, cover and catch crops, and organic soil amendments.

33: Why do you research compost and not one of the other strategies for C sequestration? You can find sources that indicate the especially promising character of compost. Compost is enhancing several soil ecosystem services, often the decay of compost is necessary and inevitable; you are mainly interested in long term C sequestration. "The soil carbon dilemma: Shall we hoard it or use it?" Janzen 2006, maybe an interesting read.

37: Is SOC accrual and SOC sequestration not redundant? This paper is very relevant (Mointe 2023); however, the message is that the contribution of SOC sequestration to climate change mitigation are most likely insignificant. Not sure if this source is justifying your research in the way you write it right now.

39: Typo, grasslands converted to grasslands? Don't think conversion of croplands to grasslands is a feasible strategy.

41: How does appropriate management look like?

45: Because the are not cultivated, no manipulation is feasible.

52: How much compost is leading to what increase in SOC stocks and where?

52-53: Source?

55-59: It seems you are about to stage your research gap and objective, but then in the next paragraph you move on with aboveground and belowground interactions, consider to improve your logic.

58-59: How do you derive at isotope labelling, provide some context and sources.

60: Why zoom out again on land management practices? You are already above mentioning that you focus is on compost.

60-66: Belowground biomass is more prone to be transformed in SOC pool, however also aboveground biomass has a significant effect on SOC stocks. Try to make this part more concise.

69: Why is water availability mentioned here?

74-75: How can compost mitigate drought?

75: You are researching compost not biochar.

78: In which direction is C allocation modified?

80-81: Belongs above to 74-75

81-82: Soil texture is not equivalent to SOM quality and quantity.

84: Is moisture retention equivalent to WHC in your context?

87-90: This argumentation lacks logic, try to split into several clear sentences.

**Methods:**

111: The normal/control management practice in this area is no fertilization?

115-116: Is this the C:N and $\delta13C$ of the maize or of the final compost? Remove ambiguity.

116-117: Summer 2019, be more specific, what month?

118: How big can I imagine the compost pile, was the composting done inside or outside in the rain? This can have significant effects on the resulting product.

119: Why did you choose this application rate? It is quite high, likely about 500 kg of N per ha.

120-121: How did you estimate the amount of C from wet weight compost?

151: How long at 960 °C?

177-222 Consider changing title to "Data analyses and statistics" since your statistics are only covered in the last part

**Results:**

253-254: Over the whole soil profile or only in the in the top 10 cm?

254-258: Confusing sentence.

259-262: Try to formulate stronger that is easier to follow that there was no significant effect on biomass after the drought treatments.

Fig. 2: change color to black for soil bulk density 0-5 cm

273-275: Make two sentences one about the effect of drought on root traits and one on the effect of compost on root traits.

300-307: Are this changes significant or not? This paragraph is hard to follow, consider rephrasing.

303: Typo, decrease in C content from 33.5 to 2.99 mg/g?

**Discussion:**

319-320: Seems logical that you find not all C back after 3 years, but also, indirect effects are possible due to increased primary production that could lead to more OM supply to the soil over time.

323-324: Obviously, if you add C to the soil, you will have more SOC, these SOC levels will decrease over time as there is no permanent stabilized C in soils.

336: Be specific, you saw an increase in % C but not in C stocks.

339-341: Also consider that these SOC changes only have a climate change mitigating effect if the C is not just transported from one place to another.

352-353: See: "The soil carbon dilemma: Shall we hoard it or use it?" Janzen 2006

**Conslusion:**

437-440: You showed results on the impact of the catenary positions, however you miss to discuss these results and only mention them briefly in the conclusion. Consider removing them from your results and other parts of the previous text or add a sufficient discussion of these results.

**General remaks:**

Be consistent with the naming of concepts.

Refer to drought or reduced precipitation, not both people might think this are two different concepts.
Same for water holding capacity, increased moisture retention and soil water retention.

---

## Author Response (AR2)

*We wish to thank the editor and the reviewers for this chance at improving our manuscript. Details of our adjustments and response to the comments follow below.*

*On behalf of all authors,*

*Daniela Guasconi*

**Report #1 (anonymous referee 3)**

**Abstract**:

14: Compared to unfertilized controls or mineral fertilized control be specific.

*Untreated controls, this was now specified in the text (L 14)*

**Introduction**:

25-26: Transfer C to the soil via root exudates and decaying aboveground and belowground organic matter.

*This was now added to the text*

29: Consider introducing the concept 4/1000 to the reader.

*We now added "initiatives aimed at increasing the C stored in soils, such as…"*

29-30: Mitigating anthropogenic GHG emissions by restoring/enhancing soil C stocks.

*The sentence was now changed to "These approaches include mitigating anthropogenic GHG emissions by restoring soil organic carbon (SOC) stocks, which can be achieved with decreasing tillage, cover crops, and with the use of soil C amendments like compost, biochar, and manure on croplands or grasslands"*

31-32: It's not only about countering tillage, it's about minimizing all agricultural management strategies that could have negative impacts on SOC stocks. i.e., no/minimal-tillage, cover and catch crops, and organic soil amendments.

*The sentence was now modified accordingly, see previous comment*

33: Why do you research compost and not one of the other strategies for C sequestration? You can find sources that indicate the especially promising character of compost. Compost is enhancing several soil ecosystem services, often the decay of compost is necessary and inevitable; you are mainly interested in long term C sequestration. "The soil carbon dilemma: Shall we hoard it or use it?" Janzen 2006, maybe an interesting read.

*Thank you for the suggested read. This experiment was not designed to test multiple strategies for C sequestration, but the initial idea was to use the same method as in Ryals and Silver (2013) and test its effect on vegetation, soil C and other variables. This was now made more explicit in the text (L 35 ).*

37: Is SOC accrual and SOC sequestration not redundant? This paper is very relevant (Mointe 2023); however, the message is that the contribution of SOC sequestration to climate change mitigation are most likely insignificant. Not sure if this source is justifying your research in the way you write it right now.

*While we argue that the concepts differ, in this case we propose to mention only sequestration and change the reference to Paltineanu et al (2024) https://doi.org/10.1016/j.catena.2024.108435*

39: Typo, grasslands converted to grasslands? Don't think conversion of croplands to grasslands is a feasible strategy.

*We argue that rather than a strategy proposed for the future, cropland conversion to grassland has occurred in many areas in Sweden in the past decades, hence the relevance to test for the effects of different management strategies (Johansson et al. 2023 - https://doi.org/10.1111/sum.13004). We propose to clarify this in the text and add the reference mentioned.*

41: How does appropriate management look like?

*We propose to reformulate to "with improved management"*

45: Because the are not cultivated, no manipulation is feasible.

*We argue that cultivation is not a necessary condition for improved management (ie pastures)*

52: How much compost is leading to what increase in SOC stocks and where?

*We added details to the text ("through an increase in plant biomass")*

52-53: Source?

*We propose to add Brown and Cotton (2011) https://doi.org/10.1080/1065657X.2011.10736983*

55-59: It seems you are about to stage your research gap and objective, but then in the next paragraph you move on with aboveground and belowground interactions, consider to improve your logic.

*Indeed, this sentence was meant to start framing the scope of our contribution. To clarify we added "Here we adopt this broad perspective and assess changes in C stocks in both soil and vegetation after C amendments." We also removed the following sentence on the use of isotopes, which only presented a methodology and not an addition concept.*

58-59: How do you derive at isotope labelling, provide some context and sources.

*We propose to remove the sentence*

60: Why zoom out again on land management practices? You are already above mentioning that you focus is on compost.

*To avoid giving the impression that we consider also other land managements, we would rephrase as "Compost amendments can impact both above- and belowground plant biomass, but these plant components contribute differently to SOC storage."*

60-66: Belowground biomass is more prone to be transformed in SOC pool, however also aboveground biomass has a significant effect on SOC stocks. Try to make this part more concise.

*This section has been rephrased (L66-74)*

69: Why is water availability mentioned here?

*Water availability was removed from this part*

74-75: How can compost mitigate drought?

*We propose to modify this section by rephrasing: "Another promising application of soil organic amendments is their use to mitigate the negative effects of drought on vegetation and soil microbial communities by increasing soil's water-holding capacity (Fischer et al. 2019; Haque et al. 2021)"*

75: You are researching compost not biochar.

*Our rationale is to test whether the effect would be similar to that of compost.*

78: In which direction is C allocation modified?

*Reduced allocation to aboveground organs, this was added to the text (L 82)*

80-81: Belongs above to 74-75

*The paragraph was edited*

81-82: Soil texture is not equivalent to SOM quality and quantity.

*We agree it is not, and we argue that water holding capacity depends on 1) texture, 2) SOM, 3) chemical composition*

84: Is moisture retention equivalent to WHC in your context?

*We refer to soil moisture retention*

87-90: This argumentation lacks logic, try to split into several clear sentences.

*We have reformulated the paragraph as: "This variability derives partly from the different physical properties of soil. However, it can also be influenced by factors such as land use history and both small- and large-scale topography (Wang et al. 2020). These complexities highlight the need for more field-based data collections—in particular under experimental conditions that combine soil amendments and drought."*

**Methods**:

111: The normal/control management practice in this area is no fertilization?

*That is correct, "no compost treatment"*

115-116: Is this the C:N and $\delta 13C$ of the maize or of the final compost? Remove ambiguity.

*These measurements refer to the final compost, as specified in the text*

116-117: Summer 2019, be more specific, what month?

*August, this was specified in the text*

118: How big can I imagine the compost pile, was the composting done inside or outside in the rain? This can have significant effects on the resulting product.

*Composting was done outside ("collected in an open field")*

119: Why did you choose this application rate? It is quite high, likely about 500 kg of N per ha.

*The method was based on "similar to the procedure described in Ryals and Silver (2013)". We followed their approach with the idea of assessing if their results are applicable to Swedish conditions.*

120-121: How did you estimate the amount of C from wet weight compost?

*This was estimated based on loss of ignition of the compost*

151: How long at 960 °C?

*2h, this was added to the text*

177-222 Consider changing title to "Data analyses and statistics" since your statistics are only covered in the last part

*This was changed as suggested*

**Results**:

253-254: Over the whole soil profile or only in the in the top 10 cm?

*Top 10cm, this was specified in the text*

254-258: Confusing sentence.

*The sentence was now split in two parts ("However, we note that mean soil C stocks in the compost-treated (ambient precipitation) plots were 6% higher in the first 15 cm, though this increase was not statistically significant. This increase is slightly higher than the percentage of compost-derived C found in that layer")*

259-262: Try to formulate stronger that is easier to follow that there was no significant effect on biomass after the drought treatments.

*We reformulated the sentence as "Experimental drought had no significant overall effect on aboveground biomass. Although biomass decreased by nearly 4% under rainout shelters (mean control plots = 642 g/m², SD = 129.23; mean drought plots = 617 g/m², SD = 180.25), this reduction was only statistically significant in compost-treated plots (P = 0.02) and not in untreated control plots."*

Fig. 2: change color to black for soil bulk density 0-5 cm

*This was changed*

273-275: Make two sentences one about the effect of drought on root traits and one on the effect of compost on root traits.

*We modified the sentence as "specific root length of coarse roots decreased under drought (P = 0.04). In contrast, after compost addition root tissue density (P = 0.02) and specific root length of all roots increased (P = 0.01)"*

300-307: Are these changes significant or not? This paragraph is hard to follow, consider rephrasing.

*They are significant, the paragraph was now rephrased: "Aboveground biomass, root biomass and soil C also differed significantly between sampling years (P < 0.05, Fig. 4, table S9). The largest change was observed in aboveground biomass, which was 53% higher in 2022 compared to 2019 (from 419.68 g m$^{-2}$, SD = 137.45 to 642.23 g m$^{-2}$, SD = 129.23). Conversely, total soil C contents and root biomass in the first 15 cm decreased by 21.5% (from 29.7 mg/g, SD = 0.73 to 23.3 mg/g, SD = 0.71) and 38.7% (from 1017.95 g m$^{-2}$, SD = 955.16 to 623.65 g m$^{-2}$, SD = 65.19), respectively."*

303: Typo, decrease in C content from 33.5 to 2.99 mg/g?

*This part was removed*

**Discussion**:

319-320: Seems logical that you find not all C back after 3 years, but also, indirect effects are possible due to increased primary production that could lead to more OM supply to the soil over time.

*We agree with the interpretation of the Reviewer, and we bring up this point in the same paragraph of the Discussion. No changes were made in response to this comment.*

323-324: Obviously, if you add C to the soil, you will have more SOC, these SOC levels will decrease over time as there is no permanent stabilized C in soils.

*Some of the compost C might be retained in the long-term, if stabilized in aggregates or on soil minerals, but we agree that this fraction is very small. In the Discussion, we argue that the chemically recalcitrant fraction of compost could remain in the soil for years, as also suggested by the C isotope analysis. No changes were made in response to this comment.*

336: Be specific, you saw an increase in % C but not in C stocks.

*This was now specified in the text*

339-341: Also consider that these SOC changes only have a climate change mitigating effect if the C is not just transported from one place to another.

*This was now specified in the text*

352-353: See: "The soil carbon dilemma: Shall we hoard it or use it?" Janzen 2006

*We added this relevant source*

**Conclusion**:

437-440: You showed results on the impact of the catenary positions, however you miss to discuss these results and only mention them briefly in the conclusion. Consider removing them from your results and other parts of the previous text or add a sufficient discussion of these results.

*The effect of catenary position is not our main focus in this contribution. Our hypotheses focus instead on the effect of our experimental treatments. The four sites thus mostly capture spatial variability in the landscape. We prefer to briefly show results about the spatial patterns for completeness, and some patterns are briefly mentioned in the first section of the Discussion, but we would not expand the Discussion on this point further. However, based on material presented in figure 3, we added more information in the Results (L 303) and Discussion (L 380).*

General remaks: Be consistent with the naming of concepts. Refer to drought or reduced precipitation, not both people might think this are two different concepts. Same for water holding capacity, increased moisture retention and soil water retention.

*We agree and have changed "precipitation reduction" to "experimental drought" where appropriate, and retained "soil water retention".*

**Report #2 (anonymous referee 2)**

Thank you for this thorough revision. Minor details that remain to be addressed are:

L170: Please provide information on scanner used (brand, model).

*The details have now been added to the manuscript (Epson Expression 10000XL flatbed scanner).*